# Unveiling patterns in human dominated landscapes through mapping the mass of US built structures

David Frantz [1,2] ✉, Franz Schug [2,3,4], Dominik Wiedenhofer [5], André Baumgart [5], Doris Virág [5], Sam Cooper[2], Camila Gómez-Medina [2], Fabian Lehmann [6], Thomas Udelhoven[7], Sebastian van der Linden [8], Patrick Hostert [2,3] & Helmut Haberl [5]

Built structures increasingly dominate the Earth's landscapes; their surging mass is currently overtaking global biomass. We here assess built structures in the conterminous US by quantifying the mass of 14 stock-building materials in eight building types and nine types of mobility infrastructures. Our high-resolution maps reveal that built structures have become 2.6 times heavier than all plant biomass across the country and that most inhabited areas are mass-dominated by buildings or infrastructure. We analyze determinants of the material intensity and show that densely built settlements have substantially lower per-capita material stocks, while highest intensities are found in sparsely populated regions due to ubiquitous infrastructures. Out-migration aggravates already high intensities in rural areas as people leave while built structures remain – highlighting that quantifying the distribution of built-up mass at high resolution is an essential contribution to understanding the biophysical basis of societies, and to inform strategies to design more resource-efficient settlements and a sustainable circular economy.

Humanity's role in changing the face of the Earth is a long-standing concern[1,2], as is the human domination of ecosystems[3]. Geologists are debating the introduction of a new geological epoch, the 'Anthropocene'[4], as humans are 'overwhelming the great forces of nature'[4,5]. In this context, the accumulation of artefacts, i.e., human-made physical objects ranging from buildings and infrastructures to machinery, is a pervasive phenomenon, locking in material and energy use, waste and greenhouse gas (GHG) emissions[6]. Variously dubbed 'manufactured capital'[7], 'technomass'[8,9], 'human-made mass'[10], 'in-use stocks'[11], or 'socioeconomic material stocks'[12], they have become a major focus of sustainability sciences in the last decade[6,13,14]. Globally,

the mass of socioeconomic material stocks now exceeds 1000 Gt, which is roughly equal to the dry-matter equivalent of all biomass on Earth[10,15]. It has been doubling roughly every 20 years, almost perfectly in line with inflation-adjusted Gross Domestic Product (GDP)[12]. In terms of mass, buildings and mobility infrastructures (here collectively called 'built structures') represent the overwhelming majority of all socioeconomic material stocks[10].

Built structures are hugely important both socially as well as ecologically. They are essential for almost all economic processes, such as production, trade, mobility, and consumption. Buildings and mobility infrastructures provide key services to societies, e.g., housing,

[1]Geoinformatics – Spatial Data Science, Trier University, Trier, Germany. [2]Geography Department, Humboldt-Universität zu Berlin, Berlin, Germany. [3]Integrated Research Institute on Transformations of Human Environment Systems (IRI THESys), Humboldt-Universität zu Berlin, Berlin, Germany. [4]SILVIS Lab, Department of Forest and Wildlife Ecology, University of Wisconsin, Madison, WI, USA. [5]Institute of Social Ecology, University of Natural Resources and Life Sciences, Vienna, Vienna, Austria. [6]Institute for Computer Science, Humboldt-Universität zu Berlin, Berlin, Germany. [7]Environmental Remote Sensing and Geoinformatics, Trier University, Trier, Germany. [8]Institute of Geography and Geology, University of Greifswald, Greifswald, Germany. ✉e-mail: david.frantz@uni-trier.de

mobility, sanitation, as well as water and energy supply[14]. Their construction now requires almost 60% of all global resource extraction[12,15]. Dynamics of built structures are a strong driver of GHG emissions[16,17] and demand for resources[18–21]. Built structures seal surfaces and increasingly cover large tracts of fertile land[22,23], structure landscapes, impede movements of species, as well as requiring increasing amounts of resource extraction[24,25], thereby exerting strong pressures on biodiversity locally and globally[26]. At the same time, natural biomass stocks play a vital role for human health, because they, for example, lower heat stress[27], while access to and time spent in green spaces improves mental health, constituting also an issue of social fairness, as witnessed during the COVID-19 pandemic[28].

Mapping these built structures at high spatial and thematic resolution is also key for understanding their role in (co-)determining future resource use emerging from mobility and housing across different settlement patterns and urban forms, as well as to assess how this reservoir of secondary materials can be re-used, re-purposed and recycled to achieve a more resource-efficient circular economy[14]. Spatially explicit insights also provide the evidence base for designing climate-change mitigation strategies targeting patterns of infrastructure and settlements[14,29] in support of the United Nations Sustainable Development Goals. Within the last decade, remote sensing-based efforts to accurately map attributes of built structures at high resolution and/or at a global scale made substantial progress regarding building density and height[30–36]. However, while globally available datasets provide important insights, their thematic depth and spatial resolution is still limited regarding structural types and material-specific compositions of built structures.

Although a global issue, material usage is not distributed evenly across the globe, and individual economies have allocated an over-proportioned share of globally extracted resources[15,32]. During the past century, the USA alone have used 16% of all materials extracted globally[37]. A better understanding of the role of spatial resource use patterns would be instrumental for identifying and locating potentials for reducing future resource use and for moving towards a more sustainable circular economy with longer lifetimes of stocks via substantially increased re-use, re-purposing, repair and recycling[14,38]. This, however, requires understanding spatial patterns of material stocks at high spatial and thematic resolution, which represents a crucial missing piece of information. Past approaches to assess material stocks in

the USA are either not spatially explicit, provided at coarse-spatial resolution, generated with incomplete geodata, or were spatially or thematically limited to single sectors[19,20,32,39–42]. A spatially explicit approach with high spatial and thematic detail have so far been prototyped by ref. 43, but only been applied to much smaller areas with more homogeneous urban forms and climate.

In this work, we combine Earth Observation data and various geodata (e.g., OpenStreetMap) with information from Industrial Ecology and technical engineering to develop a stock-driven bottom-up estimation of material stocks for the Conterminous United States in 2018 at 10 m resolution (CONUS, Supplementary Fig. 1). We advance our previous method[43] by adapting it to the high diversity of US settlement structures, environmental conditions, and construction climates. We also modify the workflow to more accurately reflect building footprints on the full spatial resolution, as well as to account for unassessed parking spaces that are expected to contain a substantial share of material stocks. Our estimation is spatially explicit, covers the entire CONUS, includes built-up structures along the entire urban-rural gradient, is based on highly complete geodata, and covers a wide range of stock types (buildings and mobility infrastructure), subtypes thereof (building and road/rail types), and material categories (e.g., concrete and steel); see Fig. 1. Based on these data, we firstly assess the geospatial distribution of the human habitat and its mass across the US, and identify locations where either mobility infrastructures, buildings, or plant stocks dominate. Secondly, we map the geospatial distribution of the material intensity of mobility and building stocks and examine which socio-economic determinants can explain these patterns.

## Results
### Material density
The total mass of built structures in the US amounts to 127 ± 5.8 Gt, i.e., ~12–13% of the global total[15] (see Supplementary Methods 3 for the computation of uncertainty). This is the equivalent of Manhattan, New York City, being buried under 966 m of solid concrete. It is almost equally divided into buildings (62.0 ± 4.7 Gt) and mobility infrastructures (64.8 ± 3.5 Gt; Fig. 2a). Ninety-eight percent of the mobility infrastructure consists of minerals, most of which (94%) are aggregates other than concrete (Fig. 2b). The composition of the building stock is more diverse with 86.2% minerals (53.4 Gt, of which 61.6% are concrete

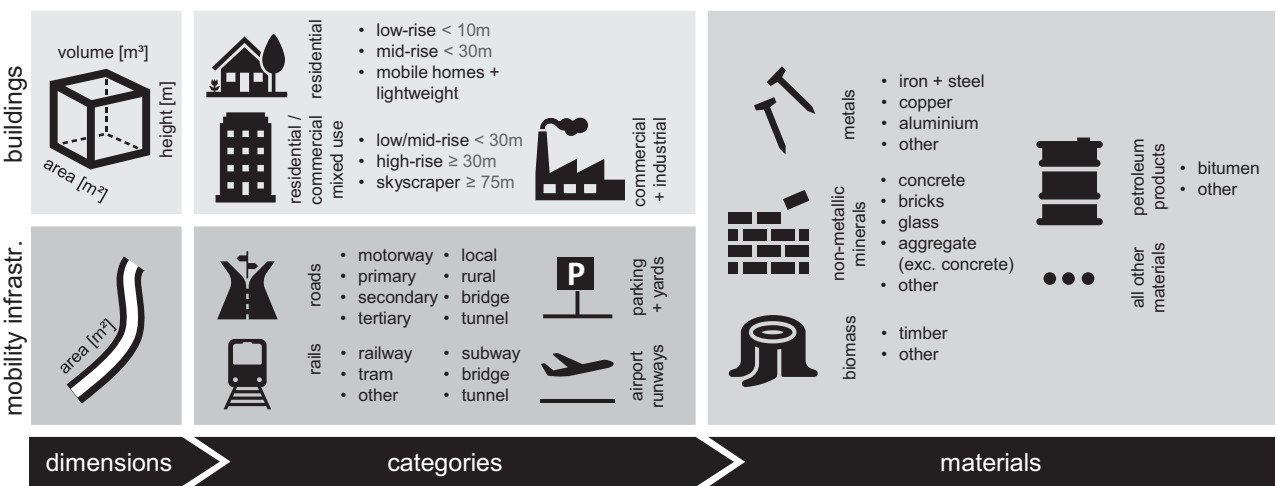

**Fig. 1 | Characteristics of built structures distinguished in our study.** We used satellite and crowdsourced geodata to map dimensions (area, height, and volume) of built structures and their categorical attributes (building types, mobility infrastructure classes) for each 10 × 10 m pixel in the entire Conterminous US. Definitions for employed categories are summarized in Supplementary Tables 4, 10, and

13. Literature-based factors of mass per m² or m³ were used to compute the mass of 14 materials in each structure category considering regionalized construction designs, hence generating a spatially explicit multidimensional description of built structures.

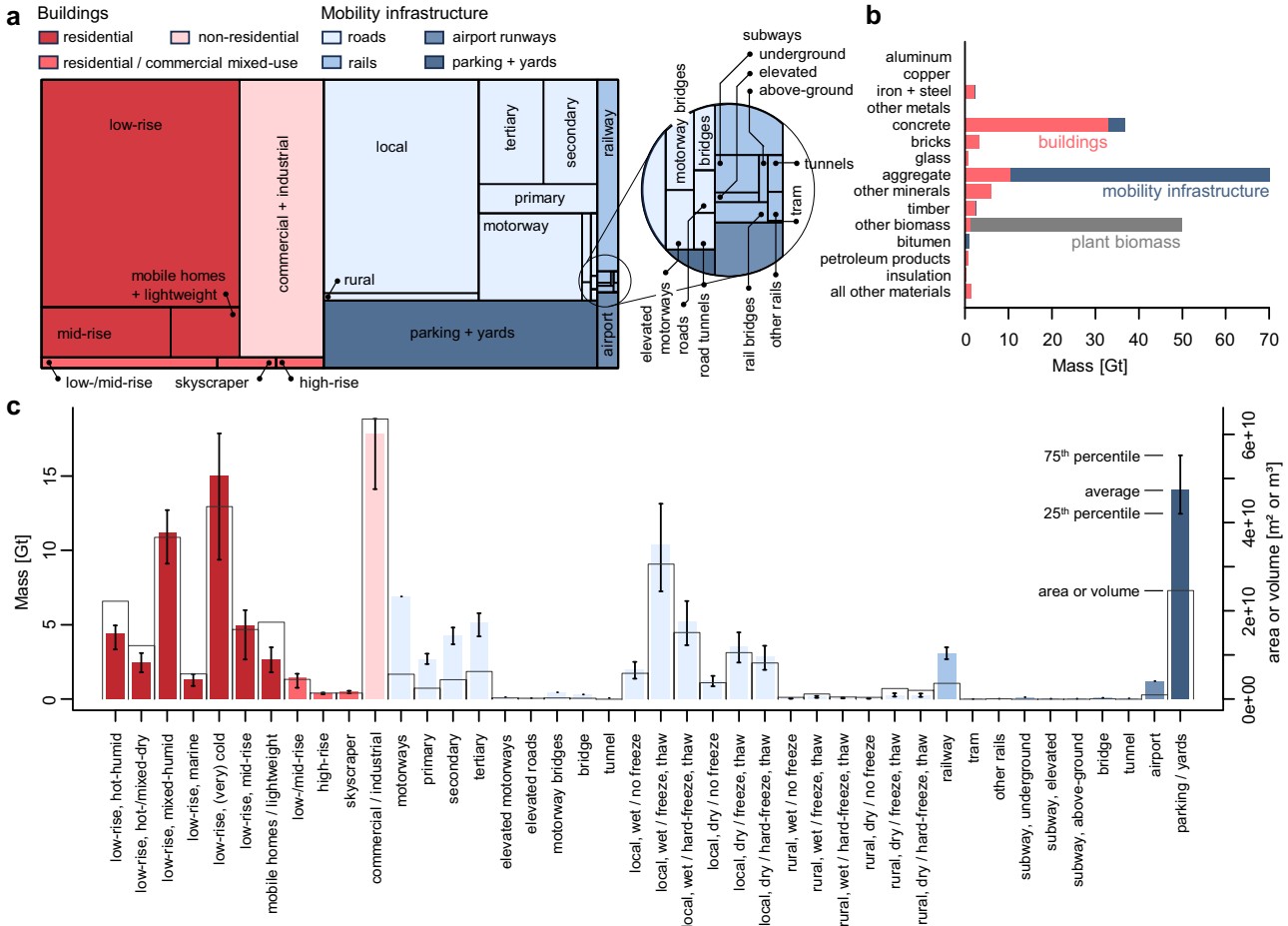

**Fig. 2 | Material stocks in the conterminous US for ca. 2018. a** relative share of material stock by categories; (**b**) material stock per material in buildings and mobility infrastructure; plant biomass is shown for reference; (**c**) uncertainty of material stock estimate (left y-axis), and total dimensions (right y-axis) for each category (including climate zone stratification); error bars represent uncertainty, which is given by the interquartile range of individual material factors, or sensitivity analysis of assumptions (see Supplemental Methods 3); the average represents the material factors used to produce the final material stock estimate; dimensions of buildings are given in m³, dimensions for mobility infrastructure in m².

and 19.6% are aggregates), 6.2% biomass, 3.6% metals, 1.5% petroleum products, and 2.6% other materials (Fig. 2b). Uncertainty arising from the assumed typologies and diversity in material composition of each structural category is presented in Fig. 2c. Uncertainty is largest in cold climates where buildings and roads need to be constructed to withstand repeated freeze and thaw cycles.

Material stocks are mapped in Fig. 3a–c. Unsurprisingly, the highest stock density (i.e., mass per unit area) prevails in urban centers, where particularly heavy structures prominently stand out, e.g., skyscrapers and subway tunnels in New York City (Fig. 3c). However, skyscrapers (0.37%) and subways (0.15%) only account for a tiny fraction of the mass of all built structures (Fig. 2a). Low-rise residential buildings (34.3 Gt) and local roads (25.2 Gt) are the heaviest subcategories overall (Fig. 2a), despite their much lower material density, due to their pervasive occurrence across almost the entire landscape.

**Dominance of human-made materials**

We use our results to map human habitats across the US (Fig. 4a) by characterizing landscapes in terms of dominance of either plant biomass (gray), buildings (red) or mobility infrastructures (blue). We here denote total stocks as the sum of these three categories, acknowledging that other, quantitatively less relevant stocks do exist, e.g., machinery, pipelines, and human, faunal or fungal biomass. We find that across the entire area of the US, the mass of built structures exceeds the 48.5 Gt of plant biomass[44] by a factor of 2.6; buildings and

mobility infrastructures each alone outweigh plants. Most of the area (61%) is dominated by stocks of built structures. Buildings prevail in urban areas, while rural agricultural areas, esp. in the Midwest and Central US, are mostly dominated by mobility infrastructures. Plants prevail in 39% of the area, especially where forest ecosystems are dominant, while biologically less productive (e.g., arid) regions are dominated by plant biomass only if they are very remote (e.g., Nevada). In 2018, only 5% of the total population lived in counties where plants outweighed built structures (Fig. 4b, n = 3,108). This finding suggests that very few people benefit from immediate access to biomass-rich green spaces that support human health and wellbeing: even half of the rural population lives in counties where more than 73% of all stocks are human-made (please refer to Supplementary Notes 1 for the employed definition of rural and urban) and fifty percent of the urban population lives in counties where plant biomass amount to 6% or less of all stocks.

The building and mobility infrastructure stock densities (t km⁻²) are highly correlated ($R^2 = 0.88$, $p < 2.2 \times 10^{-16}$, $n = 3108$, see Supplementary Methods 4), but the ratio between their respective masses varies remarkably across the US (Fig. 4a – blue vs. red). Mobility infrastructures outweigh buildings in 83% of the human-dominated areas (Fig. 4a), i.e., in those where the mass of built structures exceeds that of plant biomass. Buildings dominate only 17% of these areas, despite the fact that the total mass of buildings is almost equal to that of mobility infrastructures. A quarter of the US

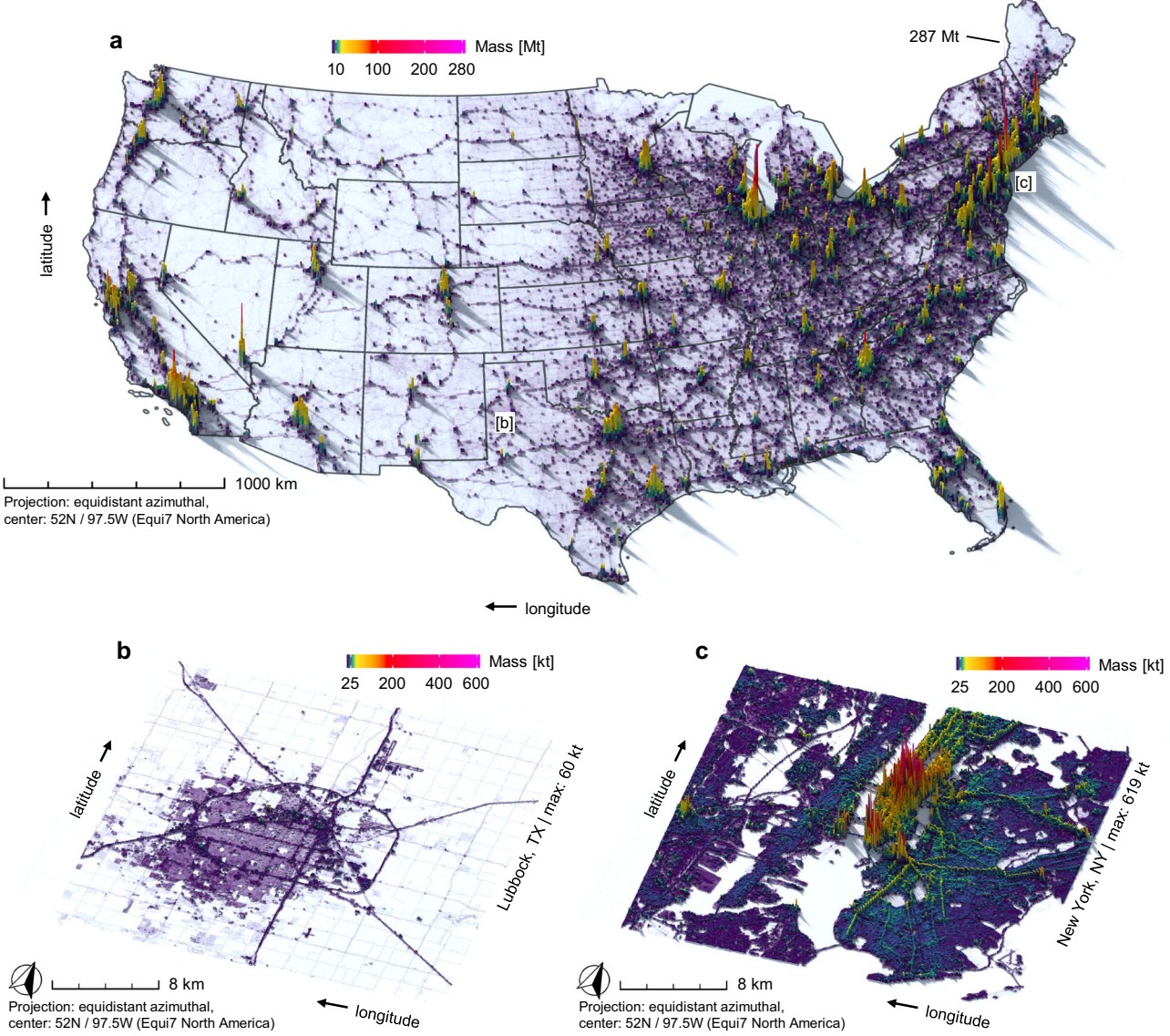

**Fig. 3 | Material stocks maps for ca. 2018. a** pseudo-3D map of the total material stock; values are in Mt per 10 × 10 km grid cell; (**b**, **c**) regional subsets and zoom-ins for New York City, NY, as well as Lubbock, TX; values are in kt per 100 × 100 m grid cell. The full 2D version of this map in units of t per 10 × 10 m grid cells is available as interactive webviewer at https://ows.geo.hu-berlin.de/webviewer/us-stocks. State boundaries were provided by the US Census Bureau.

population (26%) is living in counties where mobility infrastructures prevail, which is only 18% of the urban, yet 62% of the rural population (Fig. 4c). The urban population predominantly lives in areas where building stocks dominate.

## Material intensity of built environments

Patterns of built structures differ strongly between rural and urban areas. This becomes evident when we analyze material stocks of built structures per inhabitant (t cap$^{-1}$), here denoted as material intensity. The mass of materials piled up per person is indicative of environmental impacts, land take, and resource requirements for creation, use, and maintenance of built structures, which provide services and enable socio-economic activities[6,17,18,45,46]. The average material intensity in the US is 391 t cap$^{-1}$. We find intriguing spatial variations in material intensity: the densely populated Bronx, NY, has the lowest material intensity (90 t cap$^{-1}$), and the least-populous Loving County, TX, is identified as being the other extreme (42,691 t cap$^{-1}$). Spatial patterns, as well as county-based histograms of the material intensity of built structures are shown in Fig. 5. For

buildings, the highest material intensities (dark red) prevail in the northern Great Plains, whereas large geographic regions with low material intensities (light red) are generally found in the West, South-West, and South-East (Fig. 5a). In addition, urban counties have particularly low intensities, e.g., the urban centers along the Boston-Washington corridor. However, note that urban counties are usually small in size, thus they are not easily distinguishable in Fig. 5a, but please refer to the online version of this article where the figure's source data is provided as a table. The pattern for mobility infrastructures (Fig. 5b) differs considerably. The spatial disparity is much higher for the mobility infrastructure (595 t cap$^{-1}$ measured as interquartile range on the county level) as compared to buildings (110 t cap$^{-1}$), which is presumably because most buildings provide shelter to the local population, whereas the existence of a road does not necessarily imply that people are living nearby. Mobility infra-structures also serve agriculture, forestry and various other indus-trial activities in resource extraction, material processing and trade. Therefore, it is to be expected that those transport networks are much more spread out so that land can be accessed.

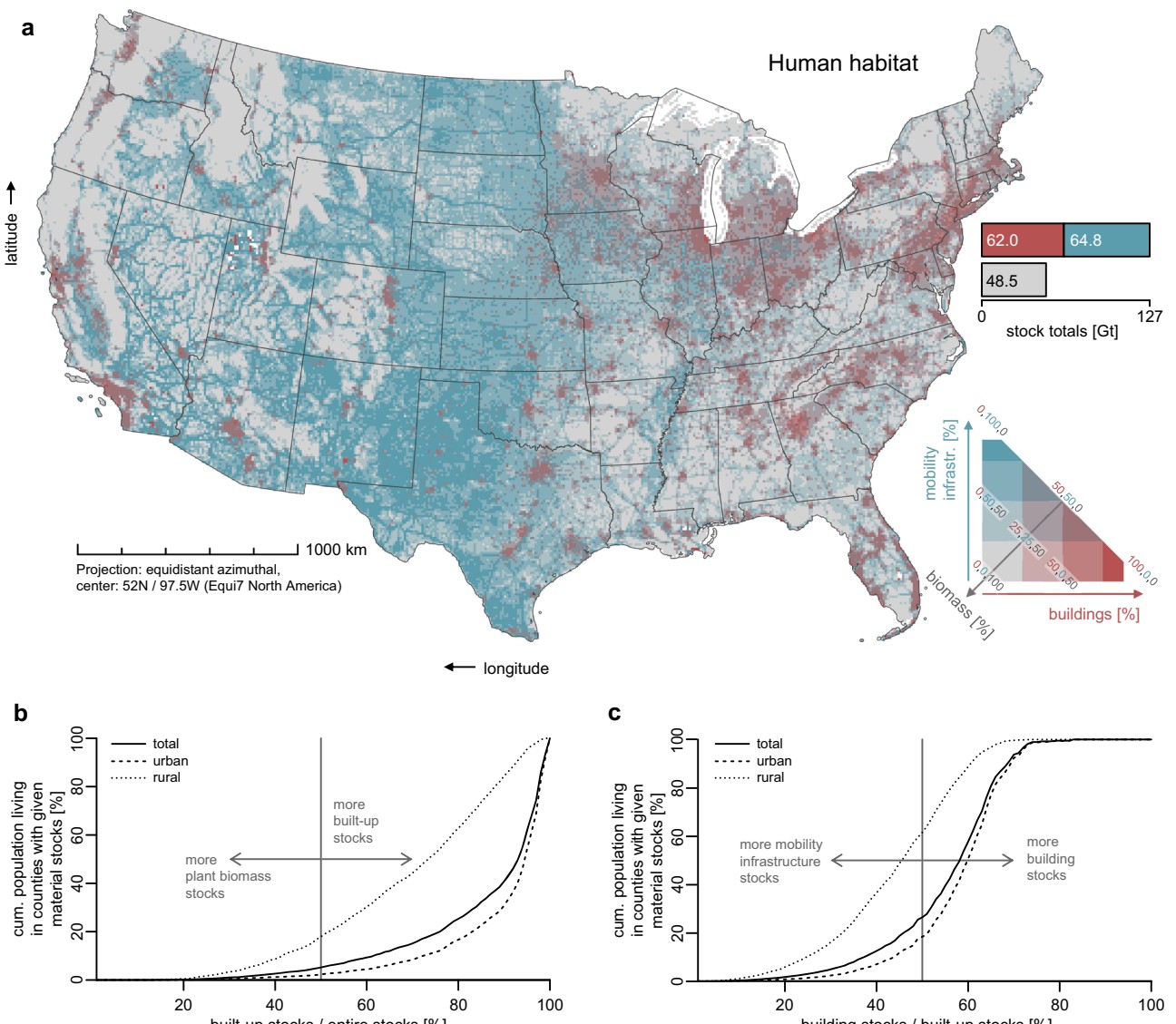

**Fig. 4 | Characterizing the 'human habitat'. a** Percentage of plant biomass (gray), building (red), and mobility infrastructure (blue) stocks relative to the entire mass of stocks within 10 × 10 km grid cells; absolute stock per component as bar chart. White pixels represent areas where no stock was found in any category, hence a percentage could not be computed. **b** Cumulative total, urban, and rural population living in counties with a given share of built-up stocks relative to the entire stocks. **c** Cumulative total, urban, and rural population living in counties with a given share of building stock relative to the total human-made material stock. Please refer to Supplementary Notes 1 for the employed definition of rural and urban. State boundaries were provided by the US Census Bureau. Plant biomass data was retrieved from ref. 44.

## Discussion

Large-scale, high-resolution mappings of societal material stocks are an increasingly hot research topic for sustainability[47]. Generating a spatially explicit, high-resolution map of material stocks across the US based on various geo- and satellite data, we find that the distribution of material stocks as well as the share of buildings vs. mobility infrastructure stocks and their relation to plant biomass are highly variable across the CONUS, and that most people are living where built-up stocks outweigh plant biomass. The rural population predominantly lives in areas with heavier mobility infrastructures while more mass is in buildings where urban populations agglomerate, respectively. Mapping material intensity in t cap⁻¹ reveals different spatial patterns and disparity for buildings and mobility infrastructures. A remarkably high variability and a pronounced spike in intensity occurs along the 100th meridian for the mobility infrastructures, whereas material intensity in buildings is more uniform across the US. These different spatial patterns suggest that the

spatial distribution of these structures have different sensitivity to population concentrations and that different socio-economic factors might be involved. While numerous factors introduce uncertainty along the workflow, our conservative bottom-up estimation of $391 \pm 18$ t cap⁻¹ is in line with previous national-scale estimates (ca. 315–430 t cap⁻¹, Supplementary Figs. 9, 10)[37,48].

This study relies on many different types of input data from different sources and of various vintage, as well as a complex processing workflow. Each input dataset and processing step adds a potential source of uncertainty, which we assess, report, and discuss; see Supplementary Table 20 for an overview of relevant supplementary items. Most potential error sources relating to the employed geodata mostly result in an underestimation of the total material stock, e.g., omission of small accessory buildings, or underprediction of tall buildings' height, although we also observe a slight overestimation for short buildings. As we combined geodata with information from industrial ecology and technical engineering to develop a spatially explicit

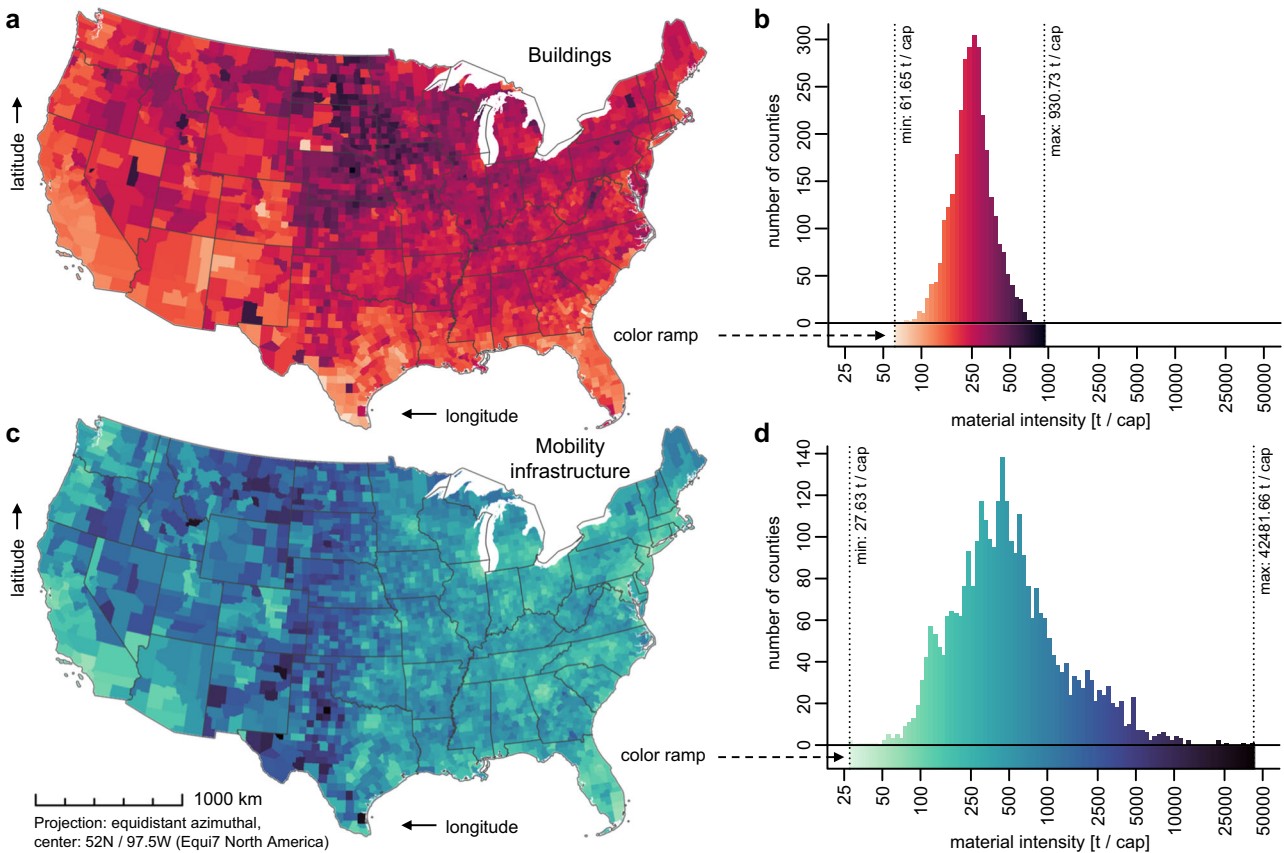

**Fig. 5 | Spatial patterns of material intensity (t cap⁻¹) of built structures.**
**a**, **b** Map and histogram of material intensity of buildings at the county level.
**c**, **d** The same for the material intensity of mobility infrastructures. Note that the data distributions of the material intensities are right-skewed, hence all data are plotted logarithmically, i.e., absolute differences in light colors are smaller than differences in dark colors. State and county boundaries were provided by the US Census Bureau.

mapping of material stocks, it is required to harmonize and align the structural types that are distinguishable in the geodata with typologies from a material perspective. Hence, the typologies assumed for each building and infrastructure category need to be generalized, which we accomplish by averaging as many sources on material composition as possible. However, there is substantial variability in published material composition per category, which we use to estimate the uncertainty of our final material stocks estimation (Fig. 2c). We find highest uncertainty in climates that necessitate more reinforced construction due to wet and cold conditions, as well as in the most abundant categories, i.e., low-rise residential buildings and local roads. The assumptions related to material stock composition itself are mostly conservative, as all our material factors are based on minimum-standard specifications according to construction codes. Actual amounts of material stocks are additionally affected, for example, by the local ground conditions potentially requiring more materials, historical construction standards for older structures potentially differing from current design specifications, as well as questions of actual compliance with specifications by construction companies. Hence, our national-scale estimation provides a reasonable estimate based on geodata streams.

Locally, deviations may nevertheless occur as general assumptions have to be made with respect to the less standardized material composition of local and rural roads: we assume a reasonable ratio of paved to non-paved surfaces as well as gravel to dirt composition for both road types, stratified by climate zones (see Methods section: Material stocks in mobility infrastructure). However, if (in a given county) gravel and dirt roads are more prevalent than assumed, estimated material stocks for these categories might be locally over-estimated to some degree, e.g., the maximum material intensity of

42,691 t cap⁻¹ in Loving County, TX could well be 33,445 t cap⁻¹ if lower bound estimates with smaller shares of paved surfaces and higher shares of dirt roads would be used for local and rural roads (according to Fig. 2c).

We find that parking surfaces amount to 11% of the total stock, or 43 t cap⁻¹ (Fig. 3a). Uncertainties exist due to our definition, where we classify all impervious areas, which are not already included in any other above-ground built-up structure, as parking-related infrastructure (Supplementary Discussion 5 and Supplementary Table 17). We find that this approach is a sound approximation as results are in good agreement with local studies: 14% of the incorporated land area of LA county[39], as well as 9.97% of the Phoenix metropolitan area[40] are covered by parking spaces, while we estimate 13.53% and 10.33%, respectively. It is also worth noting that the localized material intensity presented herein disregards that some built-up structures (partially) serve purposes that are not directly related to the local (nighttime) population. Examples are mobility infrastructure connecting distant population centers, or government facilities, as well as industrial and commercial buildings serving populations located elsewhere. A spatially-explicit understanding how stock dynamics are driven by demand elsewhere therefore constitutes an interesting new research avenue, with similarities to consumption-based environmental footprinting helping to understand distant drivers and responsibilities[49].

While individual error terms within our processing chain are well understood, we acknowledge that complex interactions can occur between different information layers, which might either aggravate or level out uncertainties. Part of the involved uncertainty can be directly used to estimate the effect on the final estimate (e.g., variability of material factors), however, the uncertainty in the geospatial products

cannot be easily translated into a final estimate of mass uncertainty. Complex simulations would be necessary to randomly permute the different aspects with respect to their identified uncertainty, and then estimate their effect on the mass estimation. This would necessitate considerable computing resources and is considered out of the scope of this paper. However, we suggest that future research should not only focus on applications of this dataset or extending it to other territories, but also on reducing uncertainty across the whole processing chain, improving the temporal alignment of input data, and assessing the combined uncertainty of the entire processing workflow. We expect that the biggest pool of uncertainty stems from the required assumptions and generalizations of the local and rural roads and would suggest prioritizing future development at this point. Especially, we consider that a reliable and spatially complete attribution of the surface type of low-grade roads would substantially reduce uncertainty as material factors could be deployed more precisely.

We consider and map building and mobility infrastructure stocks, as well as compare these to plant biomass stocks derived by ref. 44. These three stock types are inarguably the largest contributors to the entire stocks of a country. Still, other noteworthy artificial and biological stock types do exist, e.g., vehicles, industrial machinery, pipelines, electricity networks, as well as the biomass of humans, animals, or fungi, which however are quantitatively less relevant. For example, metal stocks in electricity infrastructures amount to ca. 0.45 t cap$^{-1}$ in the countries of the industrial new world (i.e., Australia, New Zealand, Canada, and the USA)[50] while material stocks of iron and steel, aluminum, copper, plastics and zinc in passenger vehicles in the USA amount to 0.89 t cap$^{-1}$ in 2015[51]. A spatially explicit inclusion of such stock types would be a valuable next step to support studies focusing on industrial sectors, material efficiency and climate change mitigation, and a more sustainable circular economy.

Independent of such considerations, intriguing and systematic spatial variations in material intensity are apparent with high intensities in the northern Great Plains and Central US for buildings and mobility infrastructures, respectively. As these are not visible in the absolute material densities, we hypothesize that the patterns might partially be explainable by socio-economic factors (Supplementary Table 17, and Supplementary Methods 5 for a multivariate analysis of potential predictors). Material intensity of buildings is most strongly associated with per-capita GDP, the percentage of urban population, household size, as well as domestic migration, birth, and international migration rates (with decreasing effect strength, Supplementary Table 18). All variables but GDP are inversely related to material intensity, which is plausible because in urban areas, housing prices are higher and land is scarce, hence more people live in multifamily houses, often with smaller flats, which require less materials per capita than single family houses – although it is noted that single-family houses are still the heaviest category in urban areas. These effects are coupled with long-term demographic changes that might cause a reinforcing feedback loop by further increasing material usage in rural areas through building vacancies and large living spaces due to population aging, whereas population gains in urban areas result in further densification or shrinkage of housing units. GDP is positively associated with material intensity, which is plausible, too, since building stocks per capita are particularly high when a high share of commercial or industrial buildings exist, which generate revenue but do not host nighttime population as counted in the census. In future studies, it might be beneficial to additionally analyze patterns of material intensity related to daytime population. The material intensity of mobility infrastructures is very strongly inversely related with population density (Supplementary Table 19). This finding reinforces the presumption that densely populated settlements require a lot less infrastructures to be supplied with mobility-related services[52,53], in contrast to scattered small villages or even single houses that require much longer roads. Other relevant variables were the 20-year domestic

migration rate, GDP, death rate, percentage of urban population, and the birth rate (with decreasing effect strength). A particularly material-intensive vertical strip can be discerned that includes the Dakotas, Nebraska, Kansas, Oklahoma, and Western Texas (Fig. 5), in which outmigration is prevalent. This suggests a strong inertia effect: while people leave, most infrastructures remain in place, even if some of them may become dysfunctional, which drives up the mass of infrastructures per capita—a phenomenon also observed elsewhere[54]. For buildings, the effect is similar, but much weaker, suggesting that buildings may be more often torn down than mobility infrastructures when people leave. GDP is positively associated again, which suggests that many roads primarily provide services to commerce and industry.

These findings have important implications for strategies towards more sustainable resource use and a more circular economy[55]. As the internationally agreed upon Sustainable Development Goals 8 & 12 state, material use needs to become much more efficient and associated environmental impacts need to be reduced rapidly. The maps presented herein can therefore serve as first order information to (a) assess potentials to utilize built structures more intensively as well as longer via re-use, repair and repurposing, (b) to (re-)designing them more efficiently, and (c) to inform the establishment of a fine-scaled network of recycling facilities to enable energy-efficient and low-emissions transport required to close material cycles with minimal environmental impacts[56]. Following the material intensity's formulation as the mass of built-up stocks divided by the population using these stocks, the intensity can either be decreased by changing population or stocks. We recommend that the built-up environment should be gradually modified to reflect the current and projected patterns of the spatial distribution of population. In areas with underutilized stocks, we suggest that the dense road network should be thinned out or re-purposed, if it is not vital for industry or commerce, while recycling potentials should be leveraged. This would both decrease costs and material input due to maintenance. In regions experiencing population growth, re-designing and adapting existing stocks, as well as increasing population density and improved recycling would contribute to reducing primary material extraction and industrial processing, as well as associated environmental impacts and emissions[57]. Renaturation could improve wildlife habitats, ecosystem resilience, and help mitigate climate change in areas where stocks are removed or re-designed[58]. In many growing areas of the U.S., where net population change is positive and material intensity is comparably high, efficient use of new stocks should be prioritized including strategic densification around key services, including access to green recreational areas and increased biomass stocks to mitigate heat stress, air pollution and improve mental health. In our study, we focused on county-level assessments, but our data would also support neighborhood-level analyses, where it is to be expected that suburban areas dominated by single-family houses with large carbon footprints[52] would fall into this category (population growth with high material intensity) and should be focus areas of sustainable densification.

We are aware that such ideas may be confronted with practical challenges. Most importantly, recovery, transportation and processing for recycling involves environmental and economic costs that need to be considered in a spatially explicit context. Our spatially resolved dataset can therefore be helpful for a first order assessment of potential sources in close geographical vicinity to areas where those materials might be reused, although siting decisions for recycling plants would certainly require additional investigation. Also, complete recycling is practically impossible, while current construction standards also limit the amount of recycled materials which can be used. Clearly, this is a complex issue, which requires in-depth and interdisciplinary research in the future, necessitating the development of an integrated geospatial model to simulate stock dynamics and spatial patterns, and associated material flows of recovered materials under different projections of population development and their

distribution, as well as policy-relevant scenarios on the re-design of existing stocks, and demand for new structures.

Overall, we find a huge range of human domination of landscapes across the conterminous US. Over one-third of its area, inhabited by only 5% of its population, is still mass-dominated by plant biomass. Most of the land, inhabited by almost all people, is dominated by built structures over plant biomass stocks. In most predominantly agricultural areas and even many other rural areas, mobility infrastructures, mostly roads, prevail in terms of mass—even if the landscape appears to be visually dominated by large areas covered by plants. Buildings outweigh mobility infrastructures only in fairly dense urban settings, where per capita material intensity of built structures, in particular mobility infrastructures, is substantially lower than elsewhere. Our results provide important insights into the structure of human-dominated landscapes, the drivers of the material intensity of different settlement patterns, as well as spatially resolved information on the potential 'urban mines' in a more resource-efficient circular economy[37]. In particular, the interrelations between demographic factors, above all domestic migration, and high vs. low material intensities along population density gradients and different settlement patterns are strikingly clear from our US-wide analysis and are important for designing strategies for climate change mitigation[52]. Developing and implementing strategies towards lowering resource use and GHG emissions in the USA can profit immensely from taking spatial patterns of built structures and their drivers into account. Beyond providing accurate and spatially explicit high-resolution estimates of material stocks distribution as a key input to socio-economic metabolism research, our study advances urban remote-sensing based research. We here establish a strong link between remote sensing imagery and socio-economic analysis, which has been identified as a strategic research goal[59]. Future research outside the United States can profit from the herein presented approach, because data availability and reference information is much lower, while the demand for information is even higher, because unique urban morphological patterns, e.g., planned vs. unplanned settlements as well as different spatial patterns and densities of settlements, can have so far unknown effects on material intensity.

## Methods

In this study, we combined Earth Observation data and various geodata with information from Industrial Ecology and technical engineering to develop a stock-driven bottom-up estimation of building and mobility infrastructure stocks for ca. 2018, i.e., the first year where the Copernicus Sentinel-2 satellite constellation was completely ramped-up for the US (see Supplementary Fig. 1 for a conceptual overview related to all following sub-sections). We mapped the material stocks for the conterminous US with high thematic (Fig. 1) and spatial detail (10 m resolution), and distinguished different stock types (buildings and mobility infrastructure), subtypes thereof (building and road/rail types), and material categories (e.g., concrete and steel). The conceptual foundation for this method was established in ref. 43, who mapped material stocks for Germany and Austria.

### Material stocks in buildings

We estimated the material stocks allocated in buildings by predicting building types and height from Earth Observation data and converting vector-based building footprints into a raster-based measurement of area. Area [m²] and height [m] were multiplied to compute aboveground building volume [m³], which was subsequently multiplied by factors for mass per volume [t m⁻³] per building type and climate zone, thus yielding material-specific mass [t] at 10 m spatial resolution.

We rasterized Microsoft building footprints (https://github.com/microsoft/USBuildingFootprints, version 1.1) to represent the building coverage [%] per pixel; The used building footprint dataset is highly complete[60]; a detailed discussion on error sources is provided in

Supplementary Discussion 1. For all raster products throughout the study, we employed a data cube structure[61] following the first tier of the EQUI7 reference grid[62], i.e., 100 km × 100 km tiles with a spatial resolution of 10 m. We further applied area correction factors for all raster-based measures of area to account for distortions of the equidistant projection (Supplementary Fig. 2), yielding the area [m²] per pixel, here, the building area [m²] per pixel.

We derived building height [m] for each built-up pixel using Sentinel-1 and Sentinel-2 data[63] for 2017 and 2018, respectively. The processing of the Earth observation data is additionally documented in Supplementary Methods 1 with predictive variables being listed in Supplementary Table 1. A support vector regression was trained with predictive features derived from the Earth observation data and building height reference information[34] from freely and openly available datasets (Supplementary Table 3), which cover a wide range of settlement types, biogeographic and climatic conditions across the US. A description and discussion of model quality is included in Supplementary Discussion 2. We additionally compiled data from the tall building database (https://www.ctbuh.org, accessed on 03.11.2020), which contains the architectural height and coordinates of buildings taller than circa 65 m. We matched the coordinates of the tall building database with building geometries from OpenStreetMap (OSM) and joined the database's height attribute with our building height layer to better accommodate for tall buildings. Aboveground building volume [m³] was computed as the product of building height and building area.

We derived the building type (see Supplementary Table 4 for classes) for each built-up pixel using Sentinel-1 and Sentinel-2 data (Supplementary Table 2). We classified the pre-dominant building type (residential, commercial and industrial, residential/commercial mixed-use, as well as lightweight buildings; Supplementary Table 5). We performed a stratified sampling approach to manually collect reference data using visual image interpretation across the US (Supplementary Table 6) and trained a random forest classifier with 1000 trees on 70% of the reference data[64]. Classification quality was evaluated on the 30% left-out data using a confusion matrix including overall accuracy, as well as class-wise user's and producer's accuracy (Supplementary Table 7 and Supplementary Discussion 3). We further refined the building type by factoring in the building height to translate the mapped classes into relevant building types from a material-specific point of view (Supplementary Table 4).

As construction standards differ between climatic zones, we employed a climate zone classification developed to aid in climate-specific building guidance[65]. We used five main climate categories, i.e., hot-humid, hot-dry/mixed-dry, mixed-humid, marine, and cold/very cold (Supplementary Table 8, Supplementary Fig. 4).

Mass factors for buildings in units of mass per volume [t m⁻³] for all building types were developed and re-estimated based on information on the materials' composition of specific, typical buildings across the US, including underground components like basements and foundations. We sourced this information from a total of 23 studies conducting life cycle assessments and material flow analysis for buildings, including structural components, underground components, as well as internal walls and furnishing (Supplementary Table 9). Preferably, studies for the US were used to derive material factors, although some foreign studies needed to be taken into account (Supplementary Table 9 and Supplementary Discussion 4). Different construction standards across climate zones are particularly relevant for residential buildings across the US, for which we developed individual factors for each climate zone defined in[65]. In the literature, mass factors for entire buildings are usually reported per building unit or per m² usable floor area. As remote-sensing information only tells us about above-ground building volumes, we converted all published information on total stocks per building to mass per volume [t m⁻³], as documented in detail in Supplementary Fig. 5 and Supplementary

Methods 2, i.e., including underground components like basements and foundations. In total, we differentiated 15 specific materials, which we published as supplementary data to this article[66], i.e., (i) metals: iron and steel, copper, aluminum, and other metals; (ii) non-metallic minerals: concrete, bricks, glass, aggregate, and other minerals; (iii) biomass-based materials: timber and other biomass-based materials such as boards; (iv) petrochemical-based materials: bitumen and other petrochemical-based materials; and (v) other materials: insulation materials and all other materials.

To estimate the mass of material stocks in buildings, we multiplied the raster-based building volume estimates [m³] with their respective mass factors [t m⁻³] – per building type, material, and if applicable per building climate zone, which yielded the mass per 10 m × 10 m pixel in tons [t].

## Material stocks in mobility infrastructure

We estimated the material stocks allocated in mobility infrastructures using crowd-sourced vector data on roads, railways, and airport surfaces, which we converted into raster-based measurement of area. The areas of remaining impervious surfaces, mostly allocated in parking spaces and yards, were derived by subtracting all other aboveground infrastructure layers used in this study from impervious cover data. The area [m²] (per mobility infrastructure type) was subsequently multiplied by material factors of mass per area [t m⁻²] per mobility infrastructure, material, and climate zone, thus yielding the mass [t] per mobility infrastructure type and material for each 10 m × 10 m pixel.

The road network was retrieved from OpenStreetMap (OSM), including information about bridges and tunnels. The OSM road network is highly complete in the US[67]. The line geometries were buffered to create a polygon dataset, wherein we defined road widths as the sum of the widths of all driving lanes, the widths of attached shoulders on both sides, and, if applicable, the width of a median strip (Supplementary Fig. 6). The buffered road polygons were subsequently transformed into a raster-based measurement of area [m²] producing images for the six main road types used, as well as for motorway and motorway link bridges, other bridges, and tunnels (see Supplementary Table 10 for details).

Mass factors for roads [t m⁻²] (Supplementary Table 11) were derived from information provided by various county-level road design manuals covering the major climatic regions of the US, and following the road definition depicted in Supplementary Fig. 6. Material factors for paved road types were developed by using weighted averages across available data sources on official minimum construction standards for each road and pavement type. We considered that paved roads are differentiated into flexible (bituminous), rigid (concrete) and composite roads[19], which were assumed to be a hybrid of flexible and rigid roads with a concrete sublayer and an asphalt surface layer[68]. The averages of mass factors for motorways, primary, secondary, and tertiary roads were weighted by the share of each pavement type in the total road length of the respective road type[69], while it was assumed that 94% of paved local roads have flexible pavement types[70].

The development of mass factors for local and rural roads required additional procedures because data quality becomes a limitation for these road types. Spatially explicit information on actual road surfaces is patchy and not consistently available for most areas of the US from OSM. Therefore, it was necessary to derive weighted and adjusted mass factors for local and rural roads to account for differences in construction, even if these differences cannot be specifically localized. Local roads can include paved roads, gravel roads, and dirt roads. Mass factors for paved local roads considered asphalted local roads. Gravel roads were defined as being constructed without asphalt or concrete pavements, and mass factors were derived from regulations on a purpose-built gravel layer[71]. Dirt roads were defined as

compacted local earth only, which in this study was interpreted as a mass factor of zero. The final mass factor for local roads was then derived by considering that 57% of local roads in the United States are paved and that 43% are unpaved[69], and the assumption that the latter consist of 25% gravel and 75% dirt roads, which was safeguarded based on selective screening of satellite photos and considering that low-grade roads are commonly more abundant than high-grade roads.

Rural roads can also include paved, gravel, and dirt roads. However, in some regions the majority of OSM tracks are classified as unknown surface. For paved rural roads, the mass factor of paved local roads was used. For regional consistency, we combined all other track categories into a single weighted rural road category and assumed that they are 50:50 gravel and dirt roads. The weighted mass for rural roads was then derived from the share of each track class in the total length of the five track classes.

Climate-specific gravel layer thicknesses[71] were further used for the development of mass factors for gravel roads as found in local and rural roads to accommodate for different design specifications due to moisture and freezing differences (Supplementary Table 12, Supplementary Fig. 7).

Mass factors for bridges and tunnels based on their respective widths were derived from the road on which the bridge or tunnel was situated (e.g., a motorway bridge is wider than a bridge for other roads). The mass factors for bridges in t m⁻² road surface were further differentiated into the actual bridge structure and the road surface on the bridge.

The railway network, including bridges, tunnels, as well as the vertical location of subways, was also retrieved from OSM and processed similarly to the road network. We defined total railway width as the width of a single-track railway including steel rails, sleepers and the underlying aggregate ballast layers (Supplementary Fig. 8), eventually yielding the area [m²] as raster images for the employed railway types (see Supplementary Table 13 for details).

Mass factors for railways (Supplementary Table 14) were primarily derived from design manuals and life cycle assessment studies and follow the definition in Supplementary Fig. 8, wherein the deepest layer of sub-ballast, which usually consists of compacted local earth, was not considered as socio-economic material stocks. As spatial information on sleeper material was not available in OSM, mass factors for railways were weighted by an average ratio of wooden to concrete sleepers reported to be 92:8 by the *Federal Railroad Administration*[72].

Airport mobility infrastructure was retrieved from OSM. Runways and taxiways were buffered according to the procedure described for roads and rails. Widths were set to 13.6 m and 9.6 m for runways and taxiways[43], respectively, unless a specific width attribute was provided by OSM. Aprons were already available as polygon data. For airport runways, the mass factor of motorways was used as a reasonable proxy regarding the comparably small area of airports (Supplementary Table 15).

Parking lot polygons were available from OSM and accordingly transformed into raster-based measurements of area. This dataset is far from complete, however, and only included large public parking spaces. Thus, we additionally included impervious surface data to include areas like private/commercial parking lots, paved yards, open sky storage, as well as any other unspecific impervious infrastructure as we expected these surfaces to hold a substantial amount of material stocks. We used the 2016 National Land Cover Database (NLCD)[73] imperviousness layer [%]. We eliminated all pixels with an impervious area <50% to mitigate smearing effects due to the necessary resampling (bilinear) from 30 m to 10 m spatial resolution. Subsequently, all area layers from the roads, railways, buildings, airport, and parking categories were subsumed (excluding underground structures, i.e., subways and tunnels) and subtracted from the impervious area, such that only impervious areas not already included in any of the other thematic layers were retained. We then subsumed parking and yard

areas (see Supplementary Discussion 5) with the OSM-extracted parking spaces into a parking and yards category. For parking lots and yards, the average of various pavement lot specific mass factors was used (Supplementary Table 15).

## Biomass stocks in plants

To compare human-made material stocks to biomass stock, we obtained global maps of above and belowground biomass carbon density for 2010[44,74]. We computed the total biomass carbon density, i.e., the sum of above- and belowground components, and converted the given values [Mg C ha$^{-1}$] to absolute masses [t] for each raster cell. We additionally employed a conversion factor of 2.0 to convert the carbon equivalent mass (C) to biomass expressed as dry matter[75].

## Data availability

The material stock data generated in this study have been deposited in the Zenodo database under accession code https://doi.org/10.5281/zenodo.8163466, and related sub-identifiers. The main material stock layers are additionally provided as interactive webviewer at https://ows.geo.hu-berlin.de/webviewer/us-stocks. The source data of printed figures are provided in the Source Data file. The raw geodata used in this study are openly available and can be obtained from the referenced resources. Source data are provided with this paper.

## Code availability

The code used for processing and analysis has been deposited in the Zenodo database under accession code https://doi.org/10.5281/zenodo.8163466, and related sub-identifiers. As an exception, preprocessed Sentinel-1 data were accessed from the commercial Earth Observation Data Centre (https://eodc.eu/), but the methodology for reproducing these is given in the supplementary methods.

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

## Acknowledgements

We thank the European Space Agency and the European Commission for providing Sentinel imagery; Clement Atzberger for supporting this study by sharing disc space on EODC; Wolfgang Wagner and EODC for providing ready-to-use Sentinel-1 data as well as support; the U.S.

Geological Survey for providing the National Land Cover Database; Microsoft for providing the building footprint data; various authorities for providing building height reference data (see Supplementary Table 3 for a full list); the Council on Tall Buildings and Urban Habitat for providing skyscraper height data; the OpenStreetMap community and Geofabrik for providing OSM data; the Dept. of Energy for providing data on building climate zones; the US Census Bureau, US Bureau of Economic Analysis for providing explanatory socio-economic variables; the US Census Bureau for providing state and county GIS boundaries; Mikhail Chester and Christopher Hoehne for sharing GIS data for intercomparison purposes. The computational results presented have been achieved (in part) using the Vienna Scientific Cluster (VSC). The work in this paper was mainly funded by the European Research Council (ERC) through the European Union's Horizon 2020 research and innovation program under grant agreement No 741950, MAT_STOCKS (D.F., F.S., D.W., A.B., D.V., C.G.-M., S.v.d.L., P.H., H.H.). Workflow development was funded by the Deutsche Forschungsgemeinschaft (DFG, German Research Foundation)—Project-ID 414984028-SFB 1404 (D.F., F.L., P.H.). The publication was funded / supported by the Open Access Fund of Universität Trier and by the German Research Foundation (DFG).

## Author contributions

Conceptualization was by D.F., H.H. and P.H. Methodology was developed by D.F., F.S., T.U., D.W., D.V., S.C., C.G.-M., P.H. and S.v.d.L. Formal Analysis was done by D.F. and F.S. Investigation was done by D.F., F.S., D.W., A.B., D.V. and T.U. Validation was done by D.F., F.S., D.W. and A.B. Visualization was by D.F., F.S., A.B. and S.v.d.L. Software was written by D.F., F.L., F.S. and T.U., Data curation was done by D.F., F.S., D.W., A.B. and D.V. Funding acquisition was by H.H. and P.H. Resources were obtained by D.F. and P.H. Project administration was by H.H. and P.H. Supervision by P.H. The original draft was written by D.F. and H.H. with review and editing carried out by H.H., P.H., F.S., D.W., D.V. and S.v.d.L.

## Funding

## Competing interests

The authors declare no competing interests.
