## [Peer Review File · Nature Communications]

Unveiling patterns in human dominated landscapes through mapping the mass of US built structuresEditorial Note: Parts of this Peer Review File have been redacted as indicated to remove third-party material where no permission to publish could be obtained.

REVIEWER COMMENTS

Reviewer #1 (Remarks to the Author):

This paper uses satellite data to provide very detailed accounting of material stocks in the United States in 2018, comparing building/transport and plant stocks. The paper is well written, interesting to read and appears to be technically sound. Contribution wise, it reproduces at a novel high level of resolution material stocks findings for the US.

The paper highlights the relative mass of human made stock versus plant stocks for different geographies and at different scale. It is not clear to me however, why I should care about the relative plant/human stock ratio? It has been known for a while that there is more human made stocks in built up areas (as could be guessed from a visual review of a city). While similar statements about the Ocean "e.g. there is more garbage in the oceans than fish" are shocking and sound very bad, I don't feel the same reaction to an on land comparison where in use stocks are not actively of themselves doing harm. I would encourage the authors to make a robust pitch for why the reader should be interested in this metric. I am very interested by the ratios between building stock and transportation stocks which are useful for thinking about what land use forms are reasonable/productive/destructive in the Anthropocene. The observation of abandoned stocks due to population movement is very interesting. The authors note this observation could be used to predict future stocks as populations move, I'd encourage them to go further and recommend where populations should go and not go given the need to limit resource use in years/decades.

There is a general omissions of horizontal infrastructure that is not transportation infrastructure, can the authors comment on how large an exclusion this is?

This paper is new in the exact question asked using this exact data, but many of the parts and methods have been used before and this is not presented as a review or a state of the art. I would encourage the authors to both more clearly note where methods are new or reference where they have been used and make an explicit argument for the added value.

Some specific comments:

line 80: How were subway tunnels detected with satellite imagery (e.g. given they are underground and can't be seen from the sky)? (later in methods this becomes clear that the subways are from OSM, up front this is not clear).

line 143: given the potentially large uncertainty in the typologies assumed for each type of building/infrastructure it would be of value to see uncertainty represented quantitatively in the results. pg 147. A few recent (and not so recent) papers have quantified the material intensity of roads and other transportation infrastructure e.g. <https://pubs.acs.org/doi/abs/10.1021/acs.est.2c05255>, <https://doi.org/10.1016/j.trd.2021.102756>, <https://doi.org/10.1016/j.resconrec.2017.08.024>, how do these findings compare to those? what is added?

line 212 Others have been published with the Sentinel 1 data asking very similar questions e.g. <https://doi.org/10.1073/pnas.2214813119>. The 2018 data sounds of higher resolution...but functionally, and given the wide uncertainties from other data inputs how much better is the newer data for outcomes/outputs? Along similar veins, functionally how much better is high resolution data compared to other approaches in the field to calculate stock, e.g. in <https://www.nature.com/articles/s41558-022-01429-y>? I'd like the authors to make a clear case that the better data gives better answers rather than spurious detail.

line 218: How was building depth/foundations/basements assessed? These are very large parts of the mass and material intensity of buildings and other structures (e.g. often up to 40% or more of buildings by mass <https://www.nature.com/articles/s41597-022-01141-8>). Figure s5 implies only above ground massing was considered?

line 282. Other papers have taken a very similar approach to road stock assessment, adding to the list

above <https://www.pnas.org/doi/abs/10.1073/pnas.2021936118>. Why repeat the process? How do the results compare to existing publications on road stocks?
line 337 Chester et al have quantified parking infrastructure a few times in different locations e.g <https://doi.org/10.1080/01944363.2015.1092879> and <https://doi.org/10.1016/j.cities.2019.02.007>

Reviewer #2 (Remarks to the Author):

Overall assessment:

The authors have developed very high resolution estimates of material stocks in buildings and mobility infrastructures in the coterminous United States. This results in a data resource which will be of great value for future research on material stocks/flows, circular economy, urban mining, embodied emissions, etc.

The quality and novelty of the data product are very clear here, but I am unconvinced about the analysis of determinants of per capita material intensities. The multivariate regressions highlight correlations of different factors with material stocks per capita, and in some cases a causal underlying mechanism appears very likely – e.g. with population density and mobility stock intensity (it is unclear why population density is not used for the model of building stock intensity). In other cases the causal connection is questionable, e.g. with griddedness and material intensity in buildings/mobility. My understanding is that more gridded street networks are more efficient than less regular street networks with more cul-de-sacs. So if future roads were built with a higher degree of griddedness, would we expect higher/lower material intensities? Prior intuition would suggest lower, but the results presented here (if representing a causal mechanism) would suggest higher. My guess is that counties with a very gridded road network, in the plains etc., tend to be in areas with very low population density, which afford large homes and lots of road length per person, and consequently high mobility and building material stocks per person, and that the results presenting positive relations between griddedness and material intensity are just correlations.

I understand that the authors are not making claims of causality. But if a multivariate regression analysis yields some interesting and expected/unexpected results, it is still the job of the authors to interpret these results, and especially to point out where correlations are likely to be spurious and not causal. Whatever the authors' interpretation and understanding of their results, I think it would be most beneficial for discussion to focus on how these data can produce analyses and recommendations which lead to lower material/resource demand in future. For future environmental impacts, changes in material stocks per person are probably less important than future inflows and outflows, as these are what determine primary resource requirements, environmental impacts, potential for material recovery, etc. If for instance, we see continued out-migration from rural areas with already high material intensity/cap but the material stock doesn't actually change, that is not really a problem for future environmental impacts, as long as there is not a continued inflow of material resources into those areas. If however road construction/maintenance continues to happen in areas with low population density and declining populations, this is a problem, because considerable material and environmental costs are incurred to benefit a small number of people.

In sum, the discussion makes some strong statements about important implications without actually spelling out what those implications or recommendations are, based on two regression analyses with some questionable interpretations. I think further depth of discussion is required regarding what are the most important determinants of material intensities, or alternatively present this as more of a data descriptor paper for a very valuable data product which can be utilised for many interesting analyses in the future. Some targeted comments on specific sections of the manuscript follow.

Comments:

Ln 22, it may be good to describe 'plants' differently, to avoid misunderstanding as 'industrial plant' etc. Consider describing as 'plant biomass' or 'plant fauna' or whatever you find most appropriate.

Ln 44-45, specifically referring to lifetime dynamics of buildings and the implications for environmental outcomes, here is another reference <https://doi.org/10.1021/acs.est.5b02333>

Ln 55, do you mean '*what*' are the dominant patterns ...?'

Fig 2a, donut/pie charts have some bad reputation for their ability to visually communicate quantities. Have you considered how this information would look on a treemap, or other alternative graph types?

Ln 100-101, it is quite obvious that by 'all stocks' you mean the total of plant biomass, buildings and mobility stocks, similar with 'entire stocks' in the x-axis of Fig 4b. However, there are also many other stocks excluded from this analysis, both biological (humans, other flora) and non-biological (industrial machinery and equipment, vehicles, etc). In future extensions of this work it would be nice to see some of these other stock types added to this type of analysis. But for now it should be explicitly stated that 'all' or 'entire' stocks refers to only the types of stocks considered here, and that there are other stocks of importance which are not included here.

Ln 107-109, you mention urban/rural counties a few times up to now. Please clarify early on how urban/rural are defined and distinguished.

Fig. 5a, we don't see the variation in building stock so much. Is it possible to show more by changing the limits of the color graph?

Ln 124-125, 'whereas low material intensities (lightred) are generally found in the West, South-West, and South-East (Fig. 5a)' – it surprises me that the dense northeast (DC-New York-Boston) is not noted as an area of low building material intensity, as it includes some of the most densely populated areas of the US. We don't actually see very well how the material intensity differs here due to the limits of the colormap (See previous comment). Is material intensity not so low here?

Ln 136, Fig. S10, can you also add comparison with <https://doi.org/10.1021/acs.est.2c05255> - which finds a material/cap value (For roads only) of 147 t/cap. See tab 'Figure S5' in their SI. This is quite close to the value you report in this study

Ln 148, can we find the 'reasonable ratio' specified somewhere?

Ln 161-162, 'fraction of urban population' needs defined as it is not immediately obvious what it refers to – is it fraction of urban population in a county with respect to annual urban population? Or fraction of population in a county that is in urban areas as opposed to rural? In the results in Table S18, is there a reason you omit population density in the model for material intensity in buildings? I wonder if that would better capture the influence of more compact/dense settlement patterns than % urban?

Ln 163, again we need the definition of urban here. Even in many census-defined 'urban' areas of the US, the dominant housing type can be single-family.

Ln 187, 'Outmigration is also highly prevalent in the Midwest' this sentence does not seem to fit with those around it.

Ln 190-192, not clear what the implications are for future resource use here, can roads in depopulated areas be taken out of service and the materials recovered? Or roads in some areas should cease to be maintained/kept in service?

Ln 259/Table S9, if you find it useful, you could add material intensities the Berrill & Hertwich (2021) study to your data collection for residential material intensities in the US. A full list of material intensities for 51 archetypes is available at

https://github.com/peterberr/US_county_HSM/blob/main/Material_Intensities/Full_arch_intensities.csv, and a regionally specific version with 30 summarised archetypes in 9 census divisions and are available at

https://github.com/peterberr/US_county_HSM/blob/main/Material_Intensities/Arch_intensities.RData

Ln 298-309, there seems to be considerable uncertainty deriving from unavoidable assumptions in the face of limited mass factor data for local and rural roads. This is especially important for local roads, as Fig 2a shows that these are the product type which make up the largest portion of mobility stock. I don't know if this is largest source of uncertainty in your workflow, but it seems to be quite an important one. It would be worth drawing attention to this sizable source of uncertainty in the main text, and pointing out relevance of this uncertainty for future studies derived from this data (e.g. on material use, or embodied emissions). Any suggestions for how future estimates of material stocks in

local roads (or other large sources of uncertainty) could become less uncertain would also be valuable. In addition – please include a short discussion/analysis on which you think are the biggest sources of uncertainty for these material estimates.

Ln 336, Table S15, have you checked whether asphalt motorways are a good proxy for airport runways? I don't have knowledge here, but would have thought that airport runways would need to be designed to withstand much greater loads than motorways. Further, the absence of any concrete from airport runways would become quite relevant for embodied emission analysis using this data. Please confirm your level of confidence on these mass factors, or improve them based on available data/literature if possible.

Reviewer #3 (Remarks to the Author):

Key results / brief summary

The contemporary debate about climate change and human-made influence on the ecosystem enforces science to assess and quantify human built material. There are international demands, e.g., UN SDGs for a sustainable treatment of our planet. Thus, the manuscript "Weighing the US: Map of Built Structures Unveils Patterns in Human-Dominated Landscapes" responds to this by an assessment of the built structures, that include mobility infrastructure and buildings, at the spatial example of the entire Conterminous United States. The authors set the built structure in relation to population and biomass and proof these accumulated structures to be 2.6 times heavier than biomass. This result is extraordinary as human kind's influence on our planet by stock material is demonstrated in an innovative way: The manuscript shows the necessity that the scientific community needs, in order to fight climate change and limited resources, that is, an interdisciplinary approach, a combination of scientific fields (remote sensing, technical engineering and social sciences). The manuscript contains very well aggregated outputs; thus, a suspenseful readable soundness is presented. Supplementary information offers a huge data background, that stands for a noteworthy work which is mirrored by the number of authors also. A wide range of calculations and huge openly accessible data source were used as a combination of high-resolution remote sensing data and geodata from the aforementioned study fields. For result presentation and visualization, the authors use maps with a continental scale, bars and histograms. An additional remarkable web viewer application offers an interactive country-wide in-depth visualization that substantiates the findings: dominance of human-made material and its spatial distribution across the entire study area; a minority of population only is living in areas with more biomass. Further, the authors proof material intensity per inhabitant, again subdivided into buildings and mobility infrastructure. I found the manuscript very insightful and informative and I think it fills a necessary gap in science. I do argue as geographer (urban morphology, EO, uncertainty mapping). In this regard I have some suggestions for improvement. For instance, some important aspects are missing: Reasons for spatial and temporal selection; some literature with respect to state of the art and uncertainties, no conclusion section and minor suggestions.

Abstract

A very well written abstract with clearly accessible aim, results, background and one sentence method. It meets the journal's standard in comparison to similar abstracts. Extent is well done with 153 words.

Introduction

Very well structured, containing paragraphs with a well-funded state of the art about global material stock. The background is transparent and aims/research questions are clearly addressed. The methodology is shortly mentioned. Considering state of the art I am missing some very important previous remote sensing works about the existence of built structures and sealing, e.g. global human

settlement layer (GHSL), global human urban footprint (GUF) as well as the recently measured World Settlement Footprint 3D building stock (WSF). Sources, for example:

-Heldens, W., Esch, T. (2015). Versiegelung – schmaler Grat zwischen Belastung und Effizienz. In: Taubenböck, H., Wurm, M., Esch, T., Dech, S. (eds) Globale Urbanisierung. Springer Spektrum, Berlin, Heidelberg. https://doi.org/10.1007/978-3-662-44841-0_13

-Pesaresi, M. et al. (2013). A Global Human Settlement Layer from optical HR/VHR RS data: concept and first results. In: IEEE Journal Of Selected Topics In Applied Earth Observations And Remote Sensing, 6, 2102 – 2131.

-Esch, T. et al. (2012). TanDEM-X mission: New perspectives for the inventory and monitoring of global settlement patterns. In: Journal of Selected Topics in Applied Earth Observation & Remote Sensing, 6, 22.

-Esch et al, (2023). World Settlement Footprint 3D - A first three-dimensional survey of the global building stock

Data, methods and analytical validity

a) Figures

All figures contain error bar explanations and they are introduced and explained in the text. The error bar explanation of fig. 5b+d concerning log can be enhanced.

-Fig3: Please provide a better resolution (image quality) for the manuscript. It's quite pixelated and unfortunately 3D bars cannot be properly distinguished in the map. However, the web application offers a better possibility.

-Fig3a+4a: A proper map should contain north arrow, scale and cartographic projection. Especially Appendix page Fig.S2 explains the error correction, thus the projection should be mentioned in the maps. Why state boundaries in fig. 3a but not in 4a?

-Fig 4a: Are lakes and rivers masked? There are white pixels in the state of Utah. Do these represent "Salt Lake"? If yes, the legend should contain a hint or put boundaries, as done in fig, 3a. Respectively the "Channel Islands of California" are represented as detached pixels.

-Fig. 5b: Please offer a better explanation of y-axis log transformation of what?

-The offered digital map with the weblink works very well and supports the two maps in the manuscript. Legend and layer options do exist.

b) Methods

-Why the temporal selection of the year 2018?

-Please explain why you chose the USA (CONUS) as spatial selection.

-OLS Regression: How were the socioeconomic variables (Appendix Table 17) normalized in order to compare resp. further calculate regression? This remains unclear to me and is not explained in the manuscript.

-Barriers and uncertainties:

1.

Line 224 and following is linked to Appendix page 4, "quality of building footprints", which I think, is not a totally sufficient barrier explanation: Relying on the mentioned method from Haberl et al., you also use random forest classifiers (Appendix Table 2) and training data set, that relies on manual visual image interpretation, right?

Also, you rely on Microsoft maps that contain computer generated building footprints. Microsoft reported precision of 99,3%. However, you note that buildings smaller 150m² are easily omitted and precision might drop under 40%. Building consolidation is your solution. So as dataset source you rely on US Building Footprints and edit and enhance the dataset.

Thus, generally building footprint measurement is, never free of uncertainties. Algorithms use manually labelled building footprints, also Microsoft did so (<https://github.com/microsoft/USBuildingFootprints>). This should be considered and highlighted in the manuscript. Literature that mentions depending uncertainties e.g., with regard to single building footprints (and heights), manual detection, used as labels for building footprints and automated processes (e.g. shadows, deviating geometries, etc.)

- Kraff, N. J., Wurm, M. & Taubenböck, H.J. (2020). Uncertainties of human perception in visual image interpretation in complex urban environments. *Journal of Selected Topics in Applied Earth Observations and Remote Sensing*. Institute of Electrical and Electronics Engineers (IEEE). Wissenschaftliche Publikation im Rahmen der Promotion beim DLR. DOI: 10.1109/JSTARS.2020.3011543
- Leichtle, T., Geiß, C., Wurm, M., Lakes, T., & Taubenböck, H. (2017). Unsupervised change detection in VHR remote sensing imagery—an object-based clustering approach in a dynamic urban environment. *International Journal of Applied Earth Observation and Geoinformation*, 54, 15-27.
- MacEachren, A. M., Robinson, A., Hopper, S., Gardner, S., Murray, R., Gahegan, M., & Hetzler, E. (2005). Visualizing geospatial information uncertainty: What we know and what we need to know. *Cartography and Geographic Information Science*, 32(3), 139-160.

2.

Further, it is mentioned you merged buildings and work with sentinel data. You might highlight this fact as single building detection needs, depending on literature, a ground sampling distance between 0.5 until 2 meters, see e.g.

- Taubenböck, H. / Dech, S. (2010): *Fernerkundung im urbanen Raum. Erdbeobachtung auf dem Weg zur Planungspraxis.* (Wissenschaftliche Buchgesellschaft). Darmstadt.
- Jacobsen, K.; Büyüksalih, G. Topographic mapping from space. In *Proceedings of the 4th Workshop of EARSeL on Remote Sensing for Developing Countries/GISDECO, Istanbul, Turkey, 4–7 June 2008*.167.
- Jensen, J. R. (2015). *Introductory digital image processing: a remote sensing perspective.* Prentice Hall Press.

c) Validity

Valid and deep-going statistic methods were used, e.g., an OLS Regression analysis was used to assess the spatial relation between building resp. mobility infrastructure and are units and a multivariate linear regression was used to set socio-economic factors in relation to the physical one. These are robust and well-known methodologies. Confusion matrix was used to describe the quality of building type prediction, the authors demonstrate transparency in their results (e.g., overall accuracy of regression with socio-economic data 0.5) and many more. The authors demonstrate a deep knowledge and findings are based on multiple deep-going statistical approaches.

Discussion

- Line 139: Uncertainty/errors remain scarce. You could add another sentence considering the above-mentioned uncertainties regarding building obtainment.
- Line 135-155: The discussion covers data and material but I am missing your output set in relation to the Introduction? Who takes a benefit from your findings?
- Line 169: "explanatory power was missing for per capita GDP". Why?
- Line 178: "might be" soundness is assumption and in discussion section inappropriate

Conclusion

Is missing and should be added easily as headline between Line 192 and 193 because the last paragraph actually sounds like a conclusion.

Appendix

- A great Appendix with lots of supplementary and necessary information to fully understand the methods used.
- In general, I found the appendix order a big chaotic and suggest ordered bullet points with clear references, e.g. line 103, "see SI" is a bit narrow.
- Fig. S5. Additionally (source 28, Haberl et al.) you can consider global building morphology indicators: Biljecki, F., & Chow, Y. S. (2022). Global building morphology indicators. *Computers*,

Environment and Urban Systems, 95, 101809.

- Fig. S.1 Pls. mark below the fig. that the flowchart was edited and the original is from Helmut Haberl, Dominik Wiedenhofer, Franz Schug, David Frantz, Doris Virág, Christoph Plutzer, Karin Gruhler, Jakob Lederer, Georg Schiller, Tomer Fishman, Maud Lanau, Andreas Gattringer, Thomas Kemper, Gang Liu, Hiroki Tanikawa, Sebastian van der Linden, and Patrick Hostert
Environmental Science & Technology 2021 55 (5), 3368-3379
DOI: 10.1021/acs.est.0c05642

Further general remarks and suggestions

- Line 102-103: R^2 calculation. You rely on your appendix, page 32 where unit and calculation is explained (per area on country level). Please outline in the text "unit area" because "unit" is not self-explanatory. In this regard Pls. also remark the appendix bullet point or page more precisely as I had to search this information. Pls. explain the noted difference between R^2 0.93 for training and 0.83 for testing

- Line 124 + 157: You mean "northern Great Plains", right? North Plains is a city.

- Line 172 + 204: You could mention the "United Nation Sustainable Development Goals" in the Introduction state of the art as scientific demand, as you rely on climate change in discussion section twice.

Originality and Significance

In the manuscript an established methodology (DOI: 10.1021/acs.est.0c05642) is used that was set up by some of the authors before. However, you apply and methodologically as well as spatially extend it to the CONUS area and use multiple calculations to underline scientific significance. A logic way to proof its global significance and continue prior research.

Previous estimations by Sreek et al. or Fishman et al, are met, yet a more deep-going and innovative approach has been done.

The findings also have significance in remote sensing with regard to settlement geography and urban morphology. In these disciplines so far, recent literature has shown that remote sensing urgently needs to be combined more often with socio-economic indicators, e.g.

- Sandborn, A. & Engstrom, R. N. (2016): Determining the Relationship Between Census Data and Spatial Features Derived From High-Resolution Imagery in Accra, Ghana, IEEE Journal of Selected Topics in Applied Earth Observations and Remote Sensing, 9(5), 1970-1977.

- Taubenböck H, Wurm M, Setiadi N, Gebert N, Roth, A, Strunz G, Birkmann J & Dech S (2009): Integrating Remote Sensing and Social Science – The correlation of urban morphology with socioeconomic parameters. IEEE-CPS Urban Remote Sensing Joint Event (JURSE), Shanghai, China. pp. 7.

Possible points of contact

Your measurements have demonstrated global building datasets. It would be very interesting to use the demonstrated methods to unveil building stock material in informal and poverty areas. Data from these areas about material stocks remain mostly undiscovered by Earth Observation so far. Thus, less obtained poverty areas also play a crucial role in your manuscript (as you mention e.g., Bronx in your article with lowest material intensity and it is well visualized in the web viewer). You respectively mentioned the relevance of the socio-economic findings in the discussion section, line 169, per capita GDP, yet that explanatory power was missing. Why? You could continue research in this area. State of the art literature that contain data barriers between formal and informal areas, rep. poverty areas, e.g.:

Kraff, N. J., Wurm, M., & Taubenböck, H. (2020). The dynamics of poor urban areas-analyzing

morphologic transformations across the globe using Earth observation data. *Cities*, 107, 102905.
Dovey, K., van Oostrum, M., Chatterjee, I., & Shafique, T. (2020). Towards a morphogenesis of informal settlements. *Habitat International*, 104, 102240.

References

The number of references is appropriate and covers most state of the art as well as methods and backgrounds. From my perspective missing parts are mentioned above (see Introduction, covering state of the art).

Thank you!

REVIEWER COMMENTS

Reviewer #1 (Remarks to the Author):

This paper uses satellite data to provide very detailed accounting of material stocks in the United States in 2018, comparing building/transport and plant stocks. The paper is well written, interesting to read and appears to be technically sound. Contribution wise, it reproduces at a novel high level of resolution material stocks findings for the US.

Thank you very much for this overall positive evaluation.

The paper highlights the relative mass of human made stock versus plant stocks for different geographies and at different scale. It is not clear to me however, why I should care about the relative plant/human stock ratio? It has been known for a while that there is more human made stocks in built up areas (as could be guessed from a visual review of a city). While similar statements about the Ocean "e.g. there is more garbage in the oceans than fish" are shocking and sound very bad, I don't feel the same reaction to an on land comparison where in use stocks are not actively of themselves doing harm. I would encourage the authors to make a robust pitch for why the reader should be interested in this metric. I am very interested by the ratios between building stock and transportation stocks which are useful for thinking about what land use forms are reasonable/productive/destructive in the Anthropocene. The observation of abandoned stocks due to population movement is very interesting. The authors note this observation could be used to predict future stocks as populations move, I'd encourage them to go further and recommend where populations should go and not go given the need to limit resource use in years/decades.

We are sorry to hear that our results on the plant biomass versus built-up stocks are not appealing to you. However, please note that the third reviewer described these results as "extraordinary" and specifically highlighted this analysis. Of course, you are correct that dominance of human-made stocks is to be expected in cities. However, we would like to emphasize that we do not only look at cities but performed a country-wide assessment. As can be seen from our results, the majority of stocks are not allocated in big cities, but distributed over the entire landscape in rather lightweight building / infrastructure types, i.e., mass through numbers. In rural areas, even landscapes that visually appear to be dominated by plants, are mass-dominated by built structures, even if these cover a smaller

fraction of the area. This is especially the case for agricultural areas in the mid-west where dense road networks outweigh the biomass that is mainly composed of crops.

Thank you for your observation regarding the building and mobility stock ratios, as well as on the population movement, which can result in individual buildings not being used anymore. However, the roads are likely still in use - but, they are used by fewer people and this also means that these streets are used more inefficiently, and some of them might actually be unnecessary (provided that longer traveling times would be acceptable). This is related to your “in use stocks not actively of themselves doing harm”. We would like to challenge this opinion. The road network of the US requires continuous maintenance and will need substantial investments in the future. Our dataset might help to prioritize areas where maintenance efforts should be focused, and potentially also identify regions where the road network is under-utilized. In this process, materials would ideally be recovered and re-used elsewhere (which would have the additional benefit of decreasing the carbon footprint of new / renewed developments and would decrease resource exploitation). In addition, we would like to note that a dense road network does in fact have negative effects on biodiversity and wildlife habitat, and reclamation would also benefit ecosystem resilience and help mitigate climate change (Selva et al. 2015).

47. Selva, N., Switalski, A., Kreft, S. & Ibsch, P. L. Why Keep Areas Road-Free? The Importance of Roadless Areas. in *Handbook of Road Ecology* 16–26 (2015). doi:10.1002/9781118568170.ch3.

The question of “where should people go?” is complex. We believe that the general goal should be to decrease material intensity, in line with the UN’s Sustainable Development Goals. As you mentioned, one potential way of influencing this metric, is to steer population development, e.g., to suggest that more people should move into the areas that are currently characterized by under-utilized structures and high material intensity. However, migration rather occurs in the opposite direction, and it seems likely that this trend continues in the future. Instead, we would rather recommend that the built-up environment should be gradually modified to reflect the current patterns and future projections of population. In our opinion, the more reasonable recommendation is to thin out the infrastructure in under-utilized areas and to adapt them to the current demand, then recover these materials and use them to focus development in areas where living space is urgently needed - but if possible, in a material-efficient way, e.g. by favoring multi-family houses over single-family houses and not by continuing the urban sprawl of single-family residential suburbs.

We have implemented a designated section in the discussion (II. 268 ff, clean version) where we describe recommendations on future resource use.

There is a general omission of horizontal infrastructure that is not transportation infrastructure, can the authors comment on how large an exclusion this is?

We have added a paragraph to the discussion about the merit of adding other types of stocks in the future. We now also give an example of published material stocks of the energy sector, where Kalt et al. (2021) find 0.45 t/cap of metal stocks in the industrial new world (Australia, New Zealand, Canada, USA). However, note that this only represents one single value for four large countries, whereas we aim at spatially mapping stocks at fine resolution within a single country. For comparison purposes, we find 7.53 t/cap of metals in buildings and mobility infrastructure. For your convenience, the new paragraph in the discussion reads as:

“We considered and mapped building and mobility infrastructure stocks, as well as compared to plant biomass stocks derived by Spawn et al.³⁸. These three stock types are inarguably the largest contributors to the entire stocks of a country. Still, other noteworthy artificial and biological stock types do exist, e.g., vehicles, industrial machinery, pipelines, electricity networks, as well as the biomass of humans, animals, or fungi. As an example, the total metal stocks in electricity infrastructures amount to ca. 0.45 t cap⁻¹ in the countries of the industrial new world (i.e., Australia, New Zealand, Canada, and the USA)⁴² and we consider that a spatially explicit addition of such stock types will be a valuable addition in the future to support studies focusing on specific industrial sectors or the circular economy in general.”

38. Spawn, S. A., Sullivan, C. C., Lark, T. J. & Gibbs, H. K. Harmonized global maps of above and belowground biomass carbon density in the year 2010. *Scientific Data* **7**, 112 (2020).

42. Kalt, G. *et al.* Material stocks in global electricity infrastructures – An empirical analysis of the power sector’s stock-flow-service nexus. *Resources, Conservation and Recycling* **173**, 105723 (2021).

This paper is new in the exact question asked using this exact data, but many of the parts and methods have been used before and this is not presented as a review or a state of the art. I would encourage the authors to both more clearly note where methods are new or reference where they have been used and make an explicit argument for the added value.

Thank you for this valuable comment. We have expanded the introduction section accordingly:

“We advanced the method recently prototyped by Haberl et al.³¹ by applying it to an area 18 times the size of the countries studied in³¹, and much more diverse in terms of settlement

structures, environmental conditions, and construction climates. We have also modified the workflow to more accurately reflect building footprints on the full spatial resolution, as well as to account for parking spaces that were previously only assessed in single locations in the US^{32,33} but are expected to hold a substantial share of resources. As compared to partially comparable studies^{19,20,28,32–35}, our estimation is spatially explicit, covers the entire CONUS, includes built-up structures along the entire urban-rural gradient, is based on geodata with high levels of completeness^{36,37}, and covers a wide range of building materials.”

19. Miatto, A., Schandl, H., Wiedenhofer, D., Krausmann, F. & Tanikawa, H. Modeling material flows and stocks of the road network in the United States 1905–2015. *Resources, Conservation and Recycling* **127**, 168–178 (2017).
20. Gregory, J. *et al.* The role of concrete in life cycle greenhouse gas emissions of US buildings and pavements. *Proceedings of the National Academy of Sciences* **118**, e2021936118 (2021).
28. Zhou, Y. *et al.* Satellite mapping of urban built-up heights reveals extreme infrastructure gaps and inequalities in the Global South. *Proceedings of the National Academy of Sciences* **119**, e2214813119 (2022).
31. Haberl, H. *et al.* High-Resolution Maps of Material Stocks in Buildings and Infrastructures in Austria and Germany. *Environ Sci Technol* **55**, 3368–3379 (2021).
32. Chester, M., Fraser, A., Matute, J., Flower, C. & Pendyala, R. Parking Infrastructure: A Constraint on or Opportunity for Urban Redevelopment? A Study of Los Angeles County Parking Supply and Growth. *Journal of the American Planning Association* **81**, 268–286 (2015).
33. Hoehne, C. G., Chester, M. V., Fraser, A. M. & King, D. A. Valley of the sun-drenched parking space: The growth, extent, and implications of parking infrastructure in Phoenix. *Cities* **89**, 186–198 (2019).
34. Rousseau, L. S. A. *et al.* Material Stock and Embodied Greenhouse Gas Emissions of Global and Urban Road Pavement. *Environ. Sci. Technol.* **56**, 18050–18059 (2022).
35. Berrill, P., Wilson, E. J. H., Reyna, J. L., Fontanini, A. D. & Hertwich, E. G. Decarbonization pathways for the residential sector in the United States. *Nature Climate Change* **12**, 712–718 (2022).
36. Barrington-Leigh, C. & Millard-Ball, A. The world's user-generated road map is more than 80% complete. *PLOS ONE* **12**, e0180698 (2017).
37. Heris, M. P., Foks, N. L., Bagstad, K. J., Troy, A. & Ancona, Z. H. A rasterized building footprint dataset for the United States. *Scientific Data* **7**, 207 (2020).

Some specific comments:

line 80: How were subway tunnels detected with satellite imagery (e.g. given they are underground and can't be seen from the sky)? (later in methods this becomes clear that the subways are from OSM, up front this is not clear).

We absolutely agree that this might be confusion. We integrate some general statements about methods into the introduction where OSM is now specifically mentioned.

line 143: given the potentially large uncertainty in the typologies assumed for each type of building/infrastructure it would be of value to see uncertainty represented quantitatively in the results.

Thank you for pointing this out. Spatially explicit mapping of material stocks necessitates some generalization on the typologies assumed for each building and infrastructure category. We derived material stock factors that are as representative as possible for the categories that we could distinguish with geodata, and derived climate-specific values where possible and reasonable to account for the diversity of the study area in terms of climate, and hence building standards. Please note that per suggestion of reviewer 2, we worked on our material factor database and ingested factors published by Berrill and Hertwich (2021). We believe that this strengthens the plausibility of our results by improving the number of data points in our database.

18. Berrill, P. & Hertwich, E. G. Material flows and GHG emissions from housing stock evolution in US counties, 2020–60. *Buildings and Cities* 2, 599–617 (2021).

In addition, we also computed low and high material stock estimates based on the individual material factors in our database. The low/high factors were derived as the 25%/75% percentiles of the individual factors and shall generate a reasonable range of possible material factors while excluding outliers. For local roads, we derived these factors based on a sensitivity analysis of assumptions. Please see the new section “Supplemental Methods 3. Uncertainty of material factors”, as well as the revised Figure 2, which now includes a bar chart for the resulting uncertainty in terms of mapped mass per category (panel c). Considering the individual ranges of the categories, the total material stock has an uncertainty of ± 5.8 Gt measured as the standard deviation of category-specific permutations of the low, average, and high estimates.

We have substantially expanded the discussion on aspects of uncertainty in the manuscript as well (ll. 180 ff, clean version).

pg 147. A few recent (and not so recent) papers have quantified the material intensity of roads and other transportation infrastructure e.g.

<https://pubs.acs.org/doi/abs/10.1021/acs.est.2c05255>,

<https://doi.org/10.1016/j.trd.2021.102756>, <https://doi.org/10.1016/j.resconrec.2017.08.024>,

how do these findings compare to those? what is added?

Thanks for these literature suggestions (also regarding your comment further below on line 282), which we are happy to add to our comparisons in the supplementary material (*Supplementary Figure 10*) - generally we find very good agreement between those estimates which are actually comparable. We do note in the supplementary information also the reasons why some of these estimates differ (different system boundaries and coverages of types of roads). Please note that *Rousseau et al.* was published shortly before this paper was submitted, yielding comparable estimates of material stocks as those presented herein; for *Miatto et al.* one of our authors was also a co-author and this study was already included in our results intercomparisons; and the third study (*Yu et al.*) is on a city in China, making it not comparable to the USA. *Gregory et al.* (comment on line 282) only considered concrete.

Furthermore, all studies were reviewed for possible addition to the material intensity database, however, for the following reasons could not be included: The *Rousseau et al.* values were already weighted with regard to surface type. Since in this study, weighted factors were used based on own assumptions and data, already weighted factors could not be added to our database as a double-weighting of factors would occur, and hence bias our material factors. Similarly, *Miatto et al.* factors could not be used since a different classification of roads based on structure (e.g., rigid vs flexible) was used, which is not transferable to OSM-derived road networks, which is based on types (e.g. motorway, local roads, etc). In this study, roads were instead differentiated based on road class hierarchy. *Yu et al.* could not be included because factors describe Chinese roads whereas here, only US (and some Canadian) sources were used.

In this study, we aimed to provide a comprehensive estimate of the spatial patterns of material stocks of buildings and infrastructure at the national level, going beyond existing studies in certain local areas (*Chester et al. 2015*, *Hoehne et al. 2017* assessed parking areas in LA county and Phoenix, respectively), incomplete geospatial data (*Miatto et al. 2017* used official network extent statistics which cover significantly fewer road kilometers compared to OSM data), intransparent and partially non-spatial input data (*Rousseau et al.* used the GRIPS dataset, which substantially underestimates road lengths, as well as the CIA's World Factbook that only holds country level, non-spatialized, roadway lengths from unknown data heritage), as well as those covering only specific materials (*Gregory et al. 2021* focused on concrete only).

We argue that our results are more complete in all these regards, i.e., considering a road network with high spatial completeness, high completeness of building footprints (please see the response to next comment), many considered materials, as well as providing a spatially explicit estimate for the whole conterminous United States at a high spatial resolution of 10m. In addition, parking spaces and similar impervious surfaces are not officially reported, yet. To this end, our developed workflow necessitates a “replication of the process” since especially our estimation of parking spaces is not independent of other built-up structures. Specifically, we use imperviousness data and subtract all known built-up structures to approximate the area of parking spaces. This can only be done using geospatial data, hence published tabular data is not sufficient for this purpose.

We are now more specific in the introduction and highlight the novelty of our study better.

19. Miatto, A., Schandl, H., Wiedenhofer, D., Krausmann, F. & Tanikawa, H. Modeling material flows and stocks of the road network in the United States 1905–2015. *Resources, Conservation and Recycling* **127**, 168–178 (2017).
20. Gregory, J. *et al.* The role of concrete in life cycle greenhouse gas emissions of US buildings and pavements. *Proceedings of the National Academy of Sciences* **118**, e2021936118 (2021).
32. Chester, M., Fraser, A., Matute, J., Flower, C. & Pendyala, R. Parking Infrastructure: A Constraint on or Opportunity for Urban Redevelopment? A Study of Los Angeles County Parking Supply and Growth. *Journal of the American Planning Association* **81**, 268–286 (2015).
33. Hoehne, C. G., Chester, M. V., Fraser, A. M. & King, D. A. Valley of the sun-drenched parking space: The growth, extent, and implications of parking infrastructure in Phoenix. *Cities* **89**, 186–198 (2019).
34. Rousseau, L. S. A. *et al.* Material Stock and Embodied Greenhouse Gas Emissions of Global and Urban Road Pavement. *Environ. Sci. Technol.* **56**, 18050–18059 (2022).
- Yu, B., Li, L., Tian, X., Yu, Q., Liu, J., & Wang, Q. (2021). Material stock quantification and environmental impact analysis of urban road systems. *Transportation Research Part D: Transport and Environment*, 93, 102756.

line 212 Others have been published with the Sentinel 1 data asking very similar questions e.g. <https://doi.org/10.1073/pnas.2214813119>. The 2018 data sounds of higher resolution...but functionally, and given the wide uncertainties from other data inputs how much better is the newer data for outcomes/outputs? Along similar veins, functionally how much better is high resolution data compared to other approaches in the field to calculate stock, e.g. in <https://www.nature.com/articles/s41558-022-01429-y>? I'd like the authors to make a clear case that the better data gives better answers rather than spurious detail.

Thanks for bringing up those studies. Compared to Zhou et al. (2022) and Berrill et al. (2022), we identify building stocks more comprehensively.

The timeframe used does not influence the spatial resolution though. Zhou et al. (2022) used Sentinel-1 data for 2015, we used Sentinel-1 and Sentinel-2 data for 2017-2018, which both have a spatial resolution of ca. 10m. However, the difference is in the employed method, both regarding masking and the derivation of the height/volume itself.

Zhou et al. *“removed non-urban areas using the urban area map from the global human settlement layer (Pesaresi et al. 2016), of which the urban areas were defined as pixels with percentages of built-up areas (e.g., roads, buildings, and parking lots) larger than 50%.”* This decision omits sparser settlement structures that are commonplace in the US in rural or suburban areas. In our study, we estimated the height for each 10m pixel covered by the Microsoft building footprint dataset, which detected footprints based on very high resolution imagery, which - as also mentioned by reviewer 3 - is a necessary prerequisite to obtain accurate building area.

This is aggravated by deriving the height/volume at an aggregated spatial resolution of 500m. Their study is based on the building height estimation of Li et al. (2020), whereas our estimation of building height is based on Frantz et al. (2021). In the latter paper, we compared our results to their dataset (for Germany). We found a good agreement for dense settlements (built-up density > 50%, i.e., the threshold employed in Zhou et al. 2022) but also found that there is no sensitivity to small buildings in sparsely populated areas in the Li et al. product, which we hypothesized is *“because the ‘aggregate-then-predict’ method hits a spectral-mixture related limit when buildings become underrepresented in a pixel. This seemingly results in a dependency of the building height prediction on building density.”* (Frantz et al. 2021)

In addition, we would argue that the height and volume definition employed in Zhou et al. (2022) does not align with the definition of building height / volume as needed for a study such as ours. They model the building height for a 500m x 500m pixel as the mean height of all underlying 30m x 30m pixels with impervious cover greater than 50%. Also, volume is computed by multiplying this height estimate with the urban area (i.e., imperviousness). That said, both the area and height components include substantial shares of non-building surfaces like streets and parking infrastructures. Similar to the two issues raised above, this introduces a bias towards dense urban centers, as it is to be expected that the building-to-other-impervious surface ratio gradually decreases along the urban-rural gradient. Conceptually, this should yield lower heights / volumes in sparser areas, which is also

reflected in Zhou et al. (2022)'s supplemental material, where they intercompared their product with our product for Germany (i.e., Frantz et al. 2021).

As can be seen in the figure below, these effects result in a substantial underestimation of the building stock in Zhou et al. (2022) as compared to our estimate. Hence, we argue that our approach provides a more complete picture, also reflected in the fact that low-rise residential buildings account for the majority of our identified building stock due to their very high prevalence throughout the whole country (55% of all buildings in terms of mass), which would be omitted to a large degree in the Zhou et al product. At this point, it also needs to be restated that even our approach still underestimates the building stock (described in Supplemental Discussion 1) because the Microsoft building footprint is less accurate for very small accessory buildings.

Please also note that the Zhou et al. (2022) methodology does not differentiate between different building types, which we consider important for deriving estimates of mass.

[redacted]

Figure: Example comparison of Zhou et al. (2022, left column) and our study (right column) for Dallas, TX. The left column depicts Zhou et al.'s building height product (note that only height was available

to download); the right column depict our building mass estimation, hence colors and data values are not directly comparable. Top to bottom: different zoom levels, note that (b) was aggregated to 500m (summed mass) for visualization purposes. Note that we verified that the area shown in (e-f) was already developed in 2015 using imagery available in Google Earth; the comparably sparsely developed area in the East is missing in (e).

Berrill et al. (2022), which is based on original estimates of building stocks by Berill and Hertwich (2021) use official census data to derive building stock estimates for the USA. However, it is known that census data is not accurate nor complete, and therefore most probably provides lower bound estimates of the actual building stock of a country. Therefore, we substantially go beyond Berrill et al. through our census-independent approach, which allows for spatially-explicit follow-up analyses.

Please also note that we included the material factors from Berill and Hertwich (2021) in our revised estimates as suggested by reviewer 2.

Berrill, P. and Hertwich, E.G., 2021. Material flows and GHG emissions from housing stock evolution in US counties, 2020–60. *Buildings and Cities*, 2(1), p.599–617. DOI: <https://doi.org/10.5334/bc.126>

Frantz, D., Schug, F., Okujeni, A., Navacchi, C., Wagner, W., van der Linden, S., & Hostert, P. (2021). National-scale mapping of building height using Sentinel-1 and Sentinel-2 time series. *Remote Sensing of Environment*, 252, 112128.

Li, X., Zhou, Y., Gong, P., Seto, K. C., & Clinton, N. (2020). Developing a method to estimate building height from Sentinel-1 data. *Remote Sensing of Environment*, 240, 111705.

Zhou, Y., Li, X., Chen, W., Meng, L., Wu, Q., Gong, P., & Seto, K. C. (2022). Satellite mapping of urban built-up heights reveals extreme infrastructure gaps and inequalities in the Global South. *Proceedings of the National Academy of Sciences*, 119(46), e2214813119.

line 218: How was building depth/foundations/basements assessed? These are very large parts of the mass and material intensity of buildings and other structures (e.g. often up to 40% or more of buildings by mass <https://www.nature.com/articles/s41597-022-01141-8>). Figure s5 implies only above ground massing was considered?

We do include the materials contained in basements and foundations through the material factors of building types, which cover the entire building as archetypes, including basements and foundations. Because remote sensing data only gives us *above-ground building volumes*, we re-estimated the building material intensity incl. basement and foundation as expressed for the entire volume of the building, to fit the “above-ground volume” definition, as documented in Supplementary Figure 5, Supplementary Methods 2, as well as in Haberl

et al. (2021). We have also clarified this point in the Methods description (Material stocks in buildings: 5. Mass factors for buildings) .

line 282. Other papers have taken a very similar approach to road stock assessment, adding to the list above <https://www.pnas.org/doi/abs/10.1073/pnas.2021936118>. Why repeat the process? How do the results compare to existing publications on road stocks?

Please see our response to your comment on line 147.

line 337 Chester et al have quantified parking infrastructure a few times in different locations e.g <https://doi.org/10.1080/01944363.2015.1092879> and <https://doi.org/10.1016/j.cities.2019.02.007>

Thank you for this literature. We inquired with Mikhail Chester and Christopher Hoehne and they were so kind to share their GIS data to allow for an intercomparison. Both studies are very well aligned with our estimate, which points towards the reliability of our approach. We added this information to the discussion section of our manuscript, too:

“Parking surfaces are an additional factor of uncertainty. Parking accounts for 43 t cap⁻¹ (Fig. 3a) and is a type of stocks previously not included in similar maps, especially not for a large spatial extent as investigated in this study. We defined all impervious areas, which were not already included in any other above-ground built-up structure, as parking-related infrastructure (Supplementary Discussion 5 and Supplementary Table 17). We found that this approach is a sound approximation as we found a good agreement with local studies: 14% of the incorporated land area of LA county ³², as well as 9.97% of the Phoenix metropolitan area ³³ are covered by parking spaces, while we estimate 13.53% and 10.33%, respectively.”

32. Chester, M., Fraser, A., Matute, J., Flower, C. & Pendyala, R. Parking Infrastructure: A Constraint on or Opportunity for Urban Redevelopment? A Study of Los Angeles County Parking Supply and Growth. *Journal of the American Planning Association* **81**, 268–286 (2015).

33. Hoehne, C. G., Chester, M. V., Fraser, A. M. & King, D. A. Valley of the sun-drenched parking space: The growth, extent, and implications of parking infrastructure in Phoenix. *Cities* **89**, 186–198 (2019).

Reviewer #2 (Remarks to the Author):

Overall assessment:

The authors have developed very high resolution estimates of material stocks in buildings and mobility infrastructures in the coterminous United States. This results in a data resource which will be of great value for future research on material stocks/flows, circular economy, urban mining, embodied emissions, etc.

Thank you very much for this evaluation. Please see our point-by-point responses to your comments, concerns, and suggestions below.

The quality and novelty of the data product are very clear here, but I am unconvinced about the analysis of determinants of per capita material intensities. The multivariate regressions highlight correlations of different factors with material stocks per capita, and in some cases a causal underlying mechanism appears very likely – e.g. with population density and mobility stock intensity (it is unclear why population density is not used for the model of building stock intensity). In other cases the causal connection is questionable, e.g. with griddedness and material intensity in buildings/mobility. My understanding is that more gridded street networks are more efficient than less regular street networks with more cul-de-sacs. So if future roads were built with a higher degree of griddedness, would we expect higher/lower material intensities? Prior intuition would suggest lower, but the results presented here (if representing a causal mechanism) would suggest higher. My guess is that counties with a very gridded road network, in the plains etc., tend to be in areas with very low population density, which afford large homes and lots of road length per person, and consequently high mobility and building material stocks per person, and that the results presenting positive relations between griddedness and material intensity are just correlations.

Thank you for this comment. We understand that parts of our regression analysis were not described in sufficient detail, we received a similar question from reviewer 3 about variables not included. We have added a more in-depth description of the method to the supplemental material (Supplemental Methods 5) and also improved on the method to draw more valid conclusions. Especially spatial autocorrelation was not considered before, but appears very important in the data employed. Left-out variables are because we performed a variable selection, which is necessary because the large sample size (3108 counties) causes all variables to be significant. The variable selection is based on a stepwise multivariate regression using the Bayesian Information Criterion (BIC) as selection criterion, and

population density was not selected by this procedure as it did not provide additional explanatory power on top of other variables.

Your comment on the griddedness is noted, and following your advice, we more closely looked into the data and noticed that the sign of the association changed randomly when running the linear regression multiple times, which is because the griddedness has a convex association with the material intensities. We still believe that griddedness might be a valuable parameter since it is related to land-use history. However, possible effects of griddedness are complex, and they would deserve more scrutiny using different approaches, which were beyond the scope here. Thus, we followed your suggestion, and removed the griddedness from our analysis in favor of enhanced interpretability of the model's results.

I understand that the authors are not making claims of causality. But if a multivariate regression analysis yields some interesting and expected/unexpected results, it is still the job of the authors to interpret these results, and especially to point out where correlations are likely to be spurious and not causal. Whatever the authors' interpretation and understanding of their results, I think it would be most beneficial for discussion to focus on how these data can produce analyses and recommendations which lead to lower material/resource demand in future. For future environmental impacts, changes in material stocks per person are probably less important than future inflows and outflows, as these are what determine primary resource requirements, environmental impacts, potential for material recovery, etc. If for instance, we see continued out-migration from rural areas with already high material intensity/cap but the material stock doesn't actually change, that is not really a problem for future environmental impacts, as long as there is not a continued inflow of material resources into those areas. If however road construction/maintenance continues to happen in areas with low population density and declining populations, this is a problem, because considerable material and environmental costs are incurred to benefit a small number of people.

Thank you very much for this suggestion. We followed your advice and formulated recommendations for future resource use based on the results of our analysis. We have implemented a designated section in the discussion for this (ll. 268 ff, clean version).

We agree that there is no additional future environmental impact if maintenance or new constructions stopped in underutilized areas, and streets would be left as they are. However, we would like to note that there still are current environmental impacts as a dense road network has negative effects on biodiversity and wildlife habitat, and reclamation would benefit ecosystem resilience and help mitigate climate change (Selva et al. 2015).

47. Selva, N., Switalski, A., Kreft, S. & Ibisch, P. L. Why Keep Areas Road-Free? The Importance of Roadless Areas. in *Handbook of Road Ecology* 16–26 (2015). doi:10.1002/9781118568170.ch3.

In addition, roads need to be maintained to guarantee traffic safety. As such, it is unclear whether maintenance could simply be ceased or whether additional measures are necessary. Thus, we would rather like to recommend that the built-up environment should be gradually modified to reflect the current patterns of population distribution and their future projections. In our opinion, the infrastructure in under-utilized areas should be adapted to the current demand, and the materials should be recovered, and re-used in areas where net population change is positive and living space is urgently needed - but if possible, in a material-efficient way, e.g. by favoring multi-family houses over single-family houses and not by continuing the urban sprawl of single-family residential suburbs. This would work towards closing the material loop in a circular economy, reduce the carbon footprint of new developments, reduce resource exploitation, and benefit wildlife habitat and help mitigate climate change.

However, we also note that our material intensity layer is based on the nighttime population. We expect that material intensity might have different spatial patterns if materials are normalized by daytime population. Hence, when deciding whether roads are underutilized or not, it should also be evaluated whether they are serving industry or commerce.

In sum, the discussion makes some strong statements about important implications without actually spelling out what those implications or recommendations are, based on two regression analyses with some questionable interpretations. I think further depth of discussion is required regarding what are the most important determinants of material intensities, or alternatively present this as more of a data descriptor paper for a very valuable data product which can be utilised for many interesting analyses in the future. Some targeted comments on specific sections of the manuscript follow.

Thank you for these suggestions. Please refer to your previous two comments. We have reworked the regression analyses, which we believe are now providing plausible results and interpretations. We also reworked the discussion of the determinants of material intensity and provided recommendations for future resource use. Please refer to the discussion, which has been substantially modified and expanded.

Comments:

Ln 22, it may be good to describe 'plants' differently, to avoid misunderstanding as 'industrial plant' etc. Consider describing as 'plant biomass' or 'plant fauna' or whatever you find most appropriate.

Point well taken. We rephrased with plant biomass throughout the manuscript. In some few occurrences, however, we only refer to plants when enough context is given.

Ln 44-45, specifically referring to lifetime dynamics of buildings and the implications for environmental outcomes, here is another reference

<https://doi.org/10.1021/acs.est.5b02333>

Thank you for the literature suggestion. We have incorporated the reference accordingly.

Ln 55, do you mean '*what* are the dominant patterns ...?'

We rephrased the research question:

"What is the geospatial distribution of the "human habitat" and its mass across the US, and where do mobility infrastructures, buildings, or plant stocks dominate?"

Fig 2a, donut/pie charts have some bad reputation for their ability to visually communicate quantities. Have you considered how this information would look on a treemap, or other alternative graph types?

The sunburst chart was primarily intended to communicate relative shares of our hierarchical data structure. However, we agree that another chart type might be better to avoid misinterpretation because of the angle-based representation. Thank you for the treemap suggestion (see new figure 2). While the plot is more busy now, it retains the ability to communicate relative shares of a hierarchically organized dataset, but additionally allows for reading the relative shares more accurately. It also allowed us to add a magnifying glass for visualizing the smaller categories.

Ln 100-101, it is quite obvious that by 'all stocks' you mean the total of plant biomass, buildings and mobility stocks, similar with 'entire stocks' in the x-axis of Fig 4b. However, there are also many other stocks excluded from this analysis, both biological (humans, other

flora) and non-biological (industrial machinery and equipment, vehicles, etc). In future extensions of this work it would be nice to see some of these other stock types added to this type of analysis. But for now it should be explicitly stated that 'all' or 'entire' stocks refers to only the types of stocks considered here, and that there are other stocks of importance which are not included here.

Thank you for this suggestion. It would indeed be interesting to augment this dataset in the future with additional stock types. We followed your advice and defined early on what our stock dataset covers, and what is meant by all or entire (ll. 123-125, clean version, bold is new):

“We use our results to map ‘human habitats’ across the US (Fig. 4a) by characterizing landscapes in terms of dominance of either plant biomass (grey), buildings (red) or mobility infrastructures (blue). **For simplicity, we here define the “entire” stocks as the sum of these three categories, however, it is noted that other important stock pools do exist, e.g., machinery, pipelines, and human, faunal or fungal biomass.**“

In the discussion, we also added a new paragraph to discuss the addition of other stock types and their potential contribution to interesting new research:

“In this study, we considered and mapped building and mobility infrastructure stocks, as well as compared to plant biomass stocks derived by ³⁸. These are inarguably the largest contributors to the entire stocks of a country. Still, other noteworthy artificial and biological stock types do exist, e.g., vehicles, industrial machinery, pipelines, electricity networks, as well as the biomass of humans, animals, or fungi. As an example, the total metal stocks in electricity infrastructures amount to ca. 0.45 t cap⁻¹ in the countries of the industrial new world (i.e., Australia, New Zealand, Canada, and the USA) ⁴². However, spatially explicit data on such stock types is usually unavailable but might be a valuable addition in the future to support future studies focusing on specific industrial sectors or the circular economy in general.”

Ln 107-109, you mention urban/rural counties a few times up to now. Please clarify early on how urban/rural are defined and distinguished.

We follow the urban and rural classification of the 2010 US census.

We have added more information in the supplemental material (Supplementary Notes 1) and refer to this section in the paper as well. For your convenience, the Supplementary section reads as:

We analyze stocks, and the relative dominance of stock categories with regards to the percentage of the urban and rural population in the US counties. We follow the urban and rural classification of the US census (2010 Census Urban and Rural Classification and Urban Area Criteria <https://www.census.gov/programs-surveys/geography/guidance/geo-areas/urban-rural/2010-urban-rural.html>, last accessed 04.07.2023):

“For the 2010 Census, an urban area will comprise a densely settled core of census tracts and/or census blocks that meet minimum population density requirements, along with adjacent territory containing non-residential urban land uses as well as territory with low population density included to link outlying densely settled territory with the densely settled core. To qualify as an urban area, the territory identified according to criteria must encompass at least 2,500 people, at least 1,500 of which reside outside institutional group quarters. The Census Bureau identifies two types of urban areas:

- *Urbanized Areas (UAs) of 50,000 or more people;*
- *Urban Clusters (UCs) of at least 2,500 and less than 50,000 people.*

Rural encompasses all population, housing, and territory not included within an urban area.”

Under the above link, the US census 2010 provides a dataset that contains the percentage of urban and rural population per county, i.e., the population living in urban or rural areas according to the definition above. The percentage of urban and rural population sums to 100%.

Fig. 5a, we don't see the variation in building stock so much. Is it possible to show more by changing the limits of the color graph?

Yes, we changed the limits of the color range to coincide with the absolute range of the data distribution (see annotation in the histogram). We also added a scale bar, cartographic projection and state boundaries as requested by reviewer 3.

Ln 124-125, 'whereas low material intensities (lightred) are generally found in the West, South-West, and South-East (Fig. 5a)' – it surprises me that the dense northeast (DC-New York-Boston) is not noted as an area of low building material intensity, as it includes some of

the most densely populated areas of the US. We don't actually see very well how the material intensity differs here due to the limits of the colormap (See previous comment). Is material intensity not so low here?

Building material intensity is indeed very low within the metropolitan areas in the northeast, e.g., Suffolk county (Boston) 129 t/cap, New York county (Manhattan, NYC) 158 t/cap, Bronx county (Bronx, NYC) 63 t/cap, Queens county (Queens, NYC) 78 t/cap, Kings county (Brooklyn, NYC) 75 t/cap, Richmond county (Staten Island, NYC) 107 t/cap, District of Columbia 145 t/cap.

However, the metropolitan counties are usually rather small, and the surrounding, comparably less densely populated counties have a higher material intensity, e.g., Bergen county, NJ 200 t/cap. In addition, owing to climatic gradients, houses are built with more materials in the North-East as compared to the South (we used climate zone-specific material factors for residential buildings), hence, material intensity is lower in the South with the same population density and building type. In addition, heavier building types are more prevalent in the North as compared to the South (see the two maps below, which depict the relative share of the area of two building categories relative to all buildings), i.e. more lightweight buildings in the South, and more residential mid-rise buildings in the North-East. This partly compensates for the density effect and results in the described large-scale geographic patterns. We formulated this more precisely in the new version (bold is new):

“For buildings, highest material intensities (dark red) prevail in the **northern Great Plains**, whereas **large geographic regions** with low material intensities (light red) are generally found in the West, South-West, and South-East (Fig. 5a). **In addition, urban counties have particularly low intensities, e.g., the urban centers along the Boston-Washington corridor.**“

Regarding color ramp, please see previous comment.

residential, mobile homes + lightweight
relative share compared to all buildings

residential, mid-rise
relative share compared to all buildings

Ln 136, Fig. S10, can you also add comparison with <https://doi.org/10.1021/acs.est.2c05255> - which finds a material/cap value (For roads only) of 147 t/cap. See tab 'Figure S5' in their SI. This is quite close to the value you report in this study

Thank you for this suggestion. Please note that *Rousseau et al.* was published shortly before this paper was submitted. We are happy to add their estimates to our comparisons in the supplementary material (*Supplementary Figure 10*), which indeed match well with our findings. We also reviewed this article for possible addition to our material intensity

database. However, their values were already weighted with regards to surface type. Since in this study, weighted factors were used based on own assumptions, already weighted factors could not be added to our database as a double-weighting of factors would occur.

Ln 148, can we find the 'reasonable ratio' specified somewhere?

This is specified in the methods section, to which we now specifically refer in the mentioned line. The corresponding methods snippets are:

"The final mass factor for local roads was then derived by considering that 57% of local roads in the United States are paved and that 43% are unpaved 59, and the assumption that the latter consist of 25% gravel and 75% dirt roads"

"Rural roads can also include paved, gravel, and dirt roads. However, in some regions the majority of OSM "tracks" are classified as unknown surface. For paved rural roads, the mass factor of paved local roads was used. For regional consistency, we combined all other track categories into a single weighted rural road category and assumed that they are 50:50 gravel and dirt roads. The weighted mass for rural roads was then derived from the share of each "track" class in the total length of the five "track" classes."

Ln 161-162, 'fraction of urban population' needs defined as it is not immediately obvious what it refers to – is it fraction of urban population in a county with respect to annual urban population? Or fraction of population in a county that is in urban areas as opposed to rural? In the results in Table S18, is there a reason you omit population density in the model for material intensity in buildings? I wonder if that would better capture the influence of more compact/dense settlement patterns than % urban?

Please see the answer to your previous comment above, as well as new Supplementary Notes 1. In short, however, the fraction of urban population is the percentage of population living in urban areas as defined by the US Census. The percentage of urban and rural population sums to 100%. We have referenced the Supplementary Notes section in Supplementary Table 17.

Please see our response to your main concerns regarding the omission of population density (in short, we used a stepwise linear model). We consider that the percentage of urban population is especially helpful as it explains situations where most stocks are allocated in a big city (so are the people that is used to normalize stocks), but the county is much larger, hence population density (normalized by area) and materials in buildings per capita do not align well.

Ln 163, again we need the definition of urban here. Even in many census-defined ‘urban’ areas of the US, the dominant housing type can be single-family.

This is true. We have further looked into this. Indeed, low-rise residential buildings (which we consider to be predominantly single-family buildings) are the heaviest category across the whole rural-urban gradient, and the material density of this category (t/km²) gradually increases with urbanity. Mid-rise residential buildings (which we consider to be predominantly multi-family buildings) as well as low-/mid-rise residential-commercial mixed use buildings (which we consider to be multi-family buildings in case of residential usage) more strongly increase towards the highest levels of urbanity. Thus, there is a general increase in single-family buildings towards high levels of urbanity, which we believe is due to more dense spatial arrangement. However, there are also more multi-family buildings in very urban areas, which overall decrease material intensity (t/cap). In addition, even the material intensity in single-family buildings decreases with urbanity, which we argue is because people are living in smaller houses or in larger households, which well compensates for the larger share of land that is being occupied by denser spatial arrangement.

We have rephrased and clarified this in the revised version:

“All variables but GDP are inversely related to material intensity, which is plausible because in urban areas, housing prices are higher and land is scarce, hence more people live in multifamily houses, often with smaller flats, which require less materials per capita than single family houses – although it is noted that single-family houses are still the heaviest category in urban areas.”

Left: Stock density (t/km²) for residential and mixed-use buildings in urban population categories.

Right: same as left but for stock intensity (t/cap)

Ln 187, 'Outmigration is also highly prevalent in the Midwest' this sentence does not seem to fit with those around it.

Thank you for the hint. We have adapted accordingly.

Ln 190-192, not clear what the implications are for future resource use here, can roads in depopulated areas be taken out of service and the materials recovered? Or roads in some areas should cease to be maintained/kept in service?

Please see our response to your general comments. We have added suggestions on the future resource used.

Ln 259/Table S9, if you find it useful, you could add material intensities the Berrill & Hertwich (2021) study to your data collection for residential material intensities in the US. A full list of material intensities for 51 archetypes is available at

https://github.com/peterberr/US_county_HSM/blob/main/Material_Intensities/Full_arch_intensities.csv, and a regionally specific version with 30 summarised archetypes in 9 census divisions and are available at

https://github.com/peterberr/US_county_HSM/blob/main/Material_Intensities/Arch_intensities.RData

Thank you for this literature suggestion, as well as the links to the data. We reviewed the paper and inspected the repository. We found that the data could well be integrated into our material factor database, and as such, we happily integrated these data into our study, which we believe strengthens the plausibility of our results by improving the number of data points in our database. Please note that including these material factors necessitated a full reprocessing and reanalysis of our data, such that all results slightly changed. The data in all linked repositories have been updated accordingly. However, we did not find substantial differences between the last version, hence the general conclusions remain the same.

Ln 298-309, there seems to be considerable uncertainty deriving from unavoidable assumptions in the face of limited mass factor data for local and rural roads. This is especially important for local roads, as Fig 2a shows that these are the product type which

make up the largest portion of mobility stock. I don't know if this is largest source of uncertainty in your workflow, but it seems to be quite an important one. It would be worth drawing attention to this sizable source of uncertainty in the main text, and pointing out relevance of this uncertainty for future studies derived from this data (e.g. on material use, or embodied emissions). Any suggestions for how future estimates of material stocks in local roads (or other large sources of uncertainty) could become less uncertain would also be valuable. In addition – please include a short discussion/analysis on which you think are the biggest sources of uncertainty for these material estimates.

You are entirely correct. We consider the local roads the single most uncertain category, especially given the high share of material stock estimated in this category. Especially “lower order” roads are hardly ever researched, and having access to more accurate material factors based on official construction guidelines, as well as their regionalization according to climate zones would definitely be an important step in the future. We now present more information regarding the uncertainty of the material factors. We computed low and high material stock estimates based on the individual material factors in our database. The low/high factors were derived as the 25%/75% percentiles of the individual factors and shall generate a reasonable range of possible material factors while excluding outliers. For local roads, we derived these factors based on a sensitivity analysis of assumptions. Please see the new section “Supplemental Methods 3. Uncertainty of material factors”, as well as the revised Figure 2, which now includes a bar chart for the resulting uncertainty in terms of mapped mass per category (panel c). A big part of the uncertainty comes from the local roads, especially in climate zones 2-3, which are wet climates with freeze-thaw circles (Eastern part of the US, excl. the Southern Coast, see Supplemental Figure 7). The uncertainty analysis yields estimates that range between 7.2-13.1 Gt and 3.9-6.6 Gt for local roads in climate zone 2 and 3, respectively.

There are also other categories that have similar uncertainty, e.g., low-rise buildings in the (very) cold climate zone (Supplemental Figure 4). However, we believe that the local roads are still more uncertain because of additional complications stemming from the employed geodata. While OpenStreetMap data are spatially accurate and complete (for the US), the attribute completeness and regional consistency thereof is still of major concern. Our local roads category is a mixture of various low order roads - which might be paved or not. Although the OSM data contains an attribute for the pavement type (or absence), this information is not routinely available, and geographic biases exist. As such, a generalization needed to be introduced, which resulted in an average material intensity that is expected to describe local roads in general. However, this can result in local deviations, e.g., if one area features more unpaved roads than the expected generalized average (see II. 199 ff, clean

version, where we discuss this with the example of Loving, TX). We expect that a complete and reliable attribution of roads with regards to pavement type (but also other attributes like lane widths, lane number, existence of sidewalks etc.) would substantially reduce uncertainty and would allow for a more targeted and detailed allocation of material stock composition and factors to different road categories, and even characteristics within the same category. We added suggestions on improvement to the discussion section (ll. 226. ff, clean version).

Ln 336, Table S15, have you checked whether asphalt motorways are a good proxy for airport runways? I don't have knowledge here, but would have thought that airport runways would need to be designed to withstand much greater loads than motorways. Further, the absence of any concrete from airport runways would become quite relevant for embodied emission analysis using this data. Please confirm your level of confidence on these mass factors, or improve them based on available data/literature if possible.

We checked again for factors of airport runways, however, the only source we were able to identify was from the Federal Aviation Agency, which reports asphalt thickness substantially lower than what we used for airport runways, thus making them even lighter. Because this source was not logical to use as airport runways (would have been half the mass of motorways), we kept motorway as a proxy. Nevertheless, before, we have only used flexible motorways as a proxy. Following information on "OurAirports.com", which describes the material composition of the majority of runways, there is also a significant share of concrete airport runways. Hence, we decided to switch to the full motorway material factor, which is a weighted average of asphalt and concrete roads, thus making the material factor for runways slightly heavier ($1.173 \text{ t/m}^2 \rightarrow 1.218 \text{ t/m}^2$ resulting in $1.17 \text{ Gt} \rightarrow 1.22 \text{ Gt}$ overall mass). Considering that airport runways represent a very small fraction of the total stock (we have specified this in the methods section), we believe that this proxy value is a reasonable choice here. For an in-depth analysis of airports, a closer look would be in order, also regarding a further disaggregation of runway categories in terms of parking spaces, aprons, and actual runways for take-off/landing, also in regards to the purpose of individual airports (e.g. freight airports vs airports mainly used for lightweight planes).

Reviewer #3 (Remarks to the Author):

Key results / brief summary

The contemporary debate about climate change and human-made influence on the ecosystem enforces science to assess and quantify human built material. There are international demands, e.g., UN SDGs for a sustainable treatment of our planet. Thus, the manuscript “Weighing the US: Map of Built Structures Unveils Patterns in Human-Dominated Landscapes” responds to this by an assessment of the built structures, that include mobility infrastructure and buildings, at the spatial example of the entire Conterminous United States. The authors set the built structure in relation to population and biomass and proof these accumulated structures to be 2.6 times heavier than biomass. This result is extraordinary as human kind’s influence on our planet by stock material is demonstrated in an innovative way: The manuscript shows the necessity that the scientific community needs, in order to fight climate change and limited resources, that is, an interdisciplinary approach, a combination of scientific fields (remote sensing, technical engineering and social sciences). The manuscript contains very well aggregated outputs; thus, a suspenseful readable soundness is presented. Supplementary information offers a huge data background, that stands for a noteworthy work which is mirrored by the number of authors also. A wide range of calculations and huge openly accessible data source were used as a combination of high-resolution remote sensing data and geodata from the aforementioned study fields. For result presentation and visualization, the authors use maps with a continental scale, bars and histograms. An additional remarkable web viewer application offers an interactive country-wide in-depth visualization that substantiates the findings: dominance of human-made material and its spatial distribution across the entire study area; a minority of population only is living in areas with more biomass. Further, the authors proof material intensity per inhabitant, again subdivided into buildings and mobility infrastructure. I found the manuscript very insightful and informative and I think it fills a necessary gap in science. I do argue as geographer (urban morphology, EO, uncertainty mapping). In this regard I have some suggestions for improvement. For instance, some important aspects are missing: Reasons for spatial and temporal selection; some literature with respect to state of the art and uncertainties, no conclusion section and minor suggestions.

Thank you very much for this kind evaluation. We implemented most of your suggestions. Please refer to the responses to the detailed feedback below.

Abstract

A very well written abstract with clearly accessible aim, results, background and one sentence method. It meets the journal's standard in comparison to similar abstracts. Extent is well done with 153 words.

Thank you for the feedback, this is much appreciated.

Introduction

Very well structured, containing paragraphs with a well-funded state of the art about global material stock. The background is transparent and aims/research questions are clearly addressed. The methodology is shortly mentioned. Considering state of the art I am missing some very important previous remote sensing works about the existence of built structures and sealing, e.g. global human settlement layer (GHSL), global human urban footprint (GUF) as well as the recently measured World Settlement Footprint 3D building stock (WSF).

Sources, for example:

-Heldens, W., Esch, T. (2015). Versiegelung – schmaler Grat zwischen Belastung und Effizienz. In: Taubenböck, H., Wurm, M., Esch, T., Dech, S. (eds) Globale Urbanisierung. Springer Spektrum, Berlin, Heidelberg. https://doi.org/10.1007/978-3-662-44841-0_13

-Pesaresi, M. et al. (2013). A Global Human Settlement Layer from optical HR/VHR RS data: concept and first results. In: IEEE Journal Of Selected Topics In Applied Earth Observations And Remote Sensing, 6, 2102 – 2131.

-Esch, T. et al. (2012). TanDEM-X mission: New perspectives for the inventory and monitoring of global settlement patterns. In: Journal of Selected Topics in Applied Earth Observation & Remote Sensing, 6, 22.

-Esch et al, (2023). World Settlement Footprint 3D - A first three-dimensional survey of the global building stock

Thanks for these helpful suggestions. We agree that a short background on existing global maps of built structures is required. We now make a short reference to the key publications of GHS and WSF in the introduction, as well as other recent papers regarding building height estimation, and clarified how our work goes beyond these studies.

It reads: "Within the last decade, remote sensing-based efforts to accurately map key attributes of built structures at high resolution and /or at a global scale made substantial progress in quantifying building density and height (Esch et al. 2012, Pesaresi et al. 2013, Li

et al. 2020a, Li et al. 2020b, Frantz et al. 2021, Esch et al. 2022, Zhou et al. 2022). However, while globally available datasets provide important insight, their thematic depth is still limited with regard to structural types and the material-specific composition of structures.”

Data, methods and analytical validity

a) Figures

All figures contain error bar explanations and they are introduced and explained in the text. The error bar explanation of fig. 5b+d concerning log can be enhanced.

We rephrased the figure caption. We have probably overcomplicated the description of the logarithmic visualization. The caption now reads:

“Fig. 5 Spatial patterns of material intensity ($t \text{ cap}^{-1}$) of built structures. (a-b) Map and histogram of material intensity of buildings at the county level. (c-d) The same for the material intensity of mobility infrastructures. Note that the data distributions of the material intensities are right-skewed, hence all data are plotted logarithmically, i.e., absolute differences in light colors are smaller than differences in dark colors.”

-Fig3: Please provide a better resolution (image quality) for the manuscript. It's quite pixelated and unfortunately 3D bars cannot be properly distinguished in the map. However, the web application offers a better possibility.

We have updated the figure with better pixel resolution. We do hope that the submission system will not compress the image again, but have also uploaded all figures as separate items in high resolution. Thank you for acknowledging the web viewer.

-Fig3a+4a: A proper map should contain north arrow, scale and cartographic projection. Especially Appendix page Fig.S2 explains the error correction, thus the projection should be mentioned in the maps. Why state boundaries in fig. 3a but not in 4a?

Thank you for these suggestions. We mostly followed your advice, though we need to rebut some of the aspects.

We initially omitted state boundaries because the lines can overlay large parts of the polygons for small counties (e.g. DC). However, we agree that it helps a lot for orientation and included state boundaries for all maps. Cartographic projection and scale bars were

added, too. Please note, however, that a textual scale indication (e.g. 1:100,000) was not used due to its incorrectness in digital maps.

We did not add north arrows to the overview maps (but to the zoom-ins in Figure 3). As an accepted cartographic convention, north arrows should only be used if maps are presented that do not follow the common north-up orientation. In addition, the chosen projection precludes the usage of a north arrow: *“Use north arrows to indicate orientation only on maps which use projections that have straight, parallel meridians, or which present areas small enough that the curvature of the earth is not noticeable. [...] the meridians converge at the north pole, so North would be indicated differently at different places on the map”* (Bunch 2002).

Bunch, M. (2002). Guidelines for Producing Cartographic Output. *Faculty of Environmental Studies, York University*. http://www.yorku.ca/gis/es3520/docs/carto_conventions.pdf

-Fig 4a: Are lakes and rivers masked? There are white pixels in the state of Utah. Do these represent “Salt Lake”? If yes, the legend should contain a hint or put boundaries, as done in fig, 3a. Respectively the “Channel Islands of California” are represented as detached pixels.

Masking was not applied. However, divide-by-zero calculations occurred in the percent-computation because all three stock categories showed 0 Gt for some 10 km x 10 km pixels in Utah (Salt Lake and surrounding desert), thus resulting in NA-values. We added this explanation to the figure caption.

Due to the addition of state boundaries, the identification of the islands should be straightforward in the updated figure.

-Fig. 5b: Please offer a better explanation of y-axes log transformation of what?

We are not entirely sure what this comment means. However, the description of the logarithmic transformation was rephrased (see the response to your first comment on figures). In Fig. 5b, this refers to the x-axis, however. The y-axis shows the frequency in terms of the number of counties (we added an additional y-axis label).

-The offered digital map with the weblink works very well and supports the two maps in the manuscript. Legend and layer options do exist.

Thank you!

b) Methods

-Why the temporal selection of the year 2018?

2018 is the first year in the US with fully ramped-up Sentinel-2 acquisitions (added to Methods description) and also represents a reasonable consensus of the various employed data sources that differ in vintage. Please note that we have added ca. 2018 to the paper wherever applicable.

-Please explain why you chose the USA (CONUS) as spatial selection.

We have expanded the introduction and now provide the justification for selecting the United States as our study area. Please note that the restriction to the *conterminous* United States was mainly due to practical reasons as a lot of additional data would have need to be crunched to include the very sparsely populated Alaska.

-OLS Regression: How were the socioeconomic variables (Appendix Table 17) normalized in order to compare resp. further calculate regression? This remains unclear to me and is not explained in the manuscript.

We understand that parts of our regression analysis was not described in sufficient detail or was hidden in the supplemental material. We have now fixed this by (i) including a more in-depth description of the method in the supplemental material (Supplemental Methods 5), (ii) improving on the structure of the supplemental material, and (iii) referencing specific parts of the supplemental material in the main paper to quickly find the relevant information. All variables were z-score normalized; population density and GDP were also log-transformed before.

-Barriers and uncertainties:

1.

Line 224 and following is linked to Appendix page 4, "quality of building footprints", which I think, is not a totally sufficient barrier explanation: Relying on the mentioned method from Haberl et al., you also use random forest classifiers (Appendix Table 2) and training data set, that relies on manual visual image interpretation, right?

We used Microsoft building footprints to derive building area, i.e., converted the polygon footprints to fractional building cover, yielding pixel values from 0-100%. The mentioned section only discusses the quality of the building footprints, i.e., the accuracy of the dataset produced by Microsoft.

In subsequent processing steps, we used Earth Observation data to map building height and type (for the pixels with building area > 0%). The discussion on building heights and type can be found in other subsections of the appendix. We understand that the presentation of the appendix was a bit chaotic before (see your comment on the appendix further below), thus we reordered and labeled sections more appropriately. In the new structure, “Quality of building height”, and “Quality of building type” directly follow the “Quality of building footprint” section. And yes, the building type prediction relied on visual image interpretation, which we now clarify in the building type section in the Methods.

Please note that, regarding building area, we deviated from the method described in Haberl et al. (2021). For Austria and Germany, we did not have access to high quality building footprint data for the entire countries, thus we approximated building area based on an empirically derived fraction of imperviousness. The approach used in this study is much more accurate in this regard and especially yields trustworthy data at the full spatial resolution of 10m (which was somewhat limited in the 2021 paper).

Also, you rely on Microsoft maps that contain computer generated building footprints. Microsoft reported precision of 99,3%. However, you note that buildings smaller 150m² are easily omitted and precision might drop under 40%. Building consolidation is your solution. So as dataset source you rely on US Building Footprints and edit and enhance the dataset.

The figure of 99.3% was directly reported by Microsoft themselves. Heris et al. 2020 performed an independent validation and they assessed the precision and recall as a function of building area. Buildings larger than 150 m² are very reliably detected, but accuracy decreases substantially for smaller buildings. We have clarified this in the supplemental material.

Building consolidation is not our solution, but another issue in this dataset (we clarified this in the supplemental material). Please see the screenshot below. This effect does not affect our estimate of the building area since we are rasterizing the polygons to generate a fractional building cover layer (0-100%), thus the individual outlines do not matter in this regard. However, we note that our prediction of building type is partially affected by this as footprint centroid density was used as one of the predictive variables.

Please note that no further editing of this dataset was applied.

[redacted]

Screenshot of the superblock consolidation issue in the Microsoft building footprint dataset for row development housing in Washington DC, overlaid on Google Earth imagery.

Thus, generally building footprint measurement is, never free of uncertainties. Algorithms use manually labelled building footprints, also Microsoft did so

(<https://github.com/microsoft/USBuildingFootprints>). This should be considered and highlighted in the manuscript. Literature that mentions depending uncertainties e.g., with regard to single building footprints (and heights), manual detection, used as labels for building footprints and automated processes (e.g. shadows, deviating geometries, etc.)

- Kraff, N. J., Wurm, M. & Taubenböck, H.J. (2020). Uncertainties of human perception in visual image interpretation in complex urban environments. Journal of Selected Topics in Applied Earth Observations and Remote Sensing. Institute of Electrical and Electronics Engineers (IEEE). Wissenschaftliche Publikation im Rahmen der Promotion beim DLR. DOI: 10.1109/JSTARS.2020.3011543

- Leichtle, T., Geiß, C., Wurm, M., Lakes, T., & Taubenböck, H. (2017). Unsupervised change detection in VHR remote sensing imagery—an object-based clustering approach in a dynamic urban environment. *International Journal of Applied Earth Observation and Geoinformation*, 54, 15-27.

- MacEachren, A. M., Robinson, A., Hopper, S., Gardner, S., Murray, R., Gahegan, M., & Hetzler, E. (2005). Visualizing geospatial information uncertainty: What we know and what we need to know. *Cartography and Geographic Information Science*, 32(3), 139-160.

You are correct that we have not mentioned data uncertainties originating from training data generation for the MS building footprint. We added a corresponding sentence in Supplementary Discussion 1. Thank you for suggesting some references!

2.

Further, it is mentioned you merged buildings and work with sentinel data. You might highlight this fact as single building detection needs, depending on literature, a ground sampling distance between 0.5 until 2 meters, see e.g.

- Taubenböck, H. / Dech, S. (2010): Fernerkundung im urbanen Raum. Erdbeobachtung auf dem Weg zur Planungspraxis. (Wissenschaftliche Buchgesellschaft). Darmstadt.

- Jacobsen, K.; Büyüksalih, G. Topographic mapping from space. In Proceedings of the 4th Workshop of EARSeL on Remote Sensing for Developing Countries/GISDECO, Istanbul, Turkey, 4–7 June 2008.167.

- Jensen, J. R. (2015). *Introductory digital image processing: a remote sensing perspective*. Prentice Hall Press.

Thank you for this comment. However, we have not performed the building detection as such with Sentinel data. We uniquely relied on Microsoft Building Footprint data (based on VHR data) for obtaining the building area. This is specified in the Methods section. We used Sentinel data to map 1) the height of buildings, and 2) the type of buildings. Building height and type information was derived for each pixel that had building coverage greater than 0 in the rasterized Microsoft dataset. Thus, we believe that the required ground sampling distance does not apply to our own workflow, but of course is a requirement for generating the building footprint dataset in the first place. Potential uncertainty in this regard is implicitly included in the accuracy assessment of the MS building footprint layer (Heris et al. 2020). While it is true that Sentinel-based mapping of building type and height is not free of uncertainty, we provide a quantitative validation of these steps in the Supplemental Material (Supplementary Discussion 2-3, Figure 3, Table 3, Table 7).

c) Validity

Valid and deep-going statistic methods were used, e.g., an OLS Regression analysis was used to assess the spatial relation between building resp. mobility infrastructure and are units and a multivariate linear regression was used to set socio-economic factors in relation to the physical one. These are robust and well-known methodologies. Confusion matrix was used to describe the quality of building type prediction, the authors demonstrate transparency in their results (e.g., overall accuracy of regression with socio-economic data 0.5) and many more. The authors demonstrate a deep knowledge and findings are based on multiple deep-going statistical approaches.

Thank you for this evaluation. Please also note that we took additional steps to statistically safeguard our regression model in response to reviewer 2's comments.

Discussion

- Line 139: Uncertainty/errors remain scarce. You could add another sentence considering the above-mentioned uncertainties regarding building obtainment.

Thank you for pointing this out. We have added more uncertainty measures to the manuscript. We especially have added an analysis regarding the variability of typologies assumed in deriving the mass factors for each category. Hence, we computed low and high material stock estimates based on the individual material factors in our database. The low/high factors were derived as the 25%/75% percentiles of the individual factors and shall generate a reasonable range of possible material factors while excluding outliers. For local roads, we derived these factors based on a sensitivity analysis of assumptions. Please see the new section "Supplemental Methods 3. Uncertainty of material factors", as well as the revised Figure 2, which now includes a bar chart for the resulting uncertainty in terms of mapped mass per category (panel c). Considering the individual ranges of the categories, the total material stock has an uncertainty of +- 5.8 Gt measured as the standard deviation of category-specific permutations of the low, average, and high estimates.

Of course, it is also correct that due to a high number of input datasets and processing steps, there are multiple potential sources of additional uncertainty in our study. We assessed, reported, and discussed the quality of all input datasets and processing steps individually in the Supplemental Material, which we now state more clearly in the Discussion section. We also collected references to all supplementary items describing uncertainty aspects in a new table (Supplementary Table 20), which makes it easier to find the relevant

information. However, we refrained from calculating final mass uncertainty based on individual error terms related to the geodata as complex interactions are possible that are not necessarily multiplicative, but might also average out. However, from the individual error assessments, we conclude that we rather underestimate the stocks, which we state qualitatively. Nevertheless, it would be beneficial in the future to specifically investigate the uncertainty aspect in much more detail, which we now also state in the manuscript. We have substantially expanded the discussion on uncertainty in the main text (ll. 180 ff, clean version).

- Line 135-155: The discussion covers data and material but I am missing your output set in relation to the Introduction? Who takes a benefit from your findings?

We have substantially expanded the discussion with regards to many key aspects raised by you and the other two reviewers. We have formulated specific recommendations for the future material usage of the US, and have also added some sentences at the end of the discussion regarding potential future studies inside and outside of the US.

- Line 169: “explanatory power was missing for per capita GDP”. Why?

We performed a variable selection, which was necessary because the large sample size (3108 counties) causes all variables to be significant. The variable selection is based on a stepwise multivariate regression using the Bayesian Information Criterion (BIC) as selection criterion. This information was already included in the supplemental material, but as described earlier, we have now expanded on the methods description, and have also added this information to the table caption of the socio-economic variables.

Specifically regarding GDP, we have further looked into the issue and found that we did not normalize GDP by population. As such, GDP was highly correlated with population density and was not selected by the model. In the new version, we have corrected this, and GDP is now included in both multivariate regressions.

- Line 178: “might be” soundness is assumption and in discussion section inappropriate

Due to suggestions from reviewer 2, this sentence is no longer part of the main text.

Conclusion

Is missing and should be added easily as headline between Line 192 and 193 because the last paragraph actually sounds like a conclusion.

We agree. However, the journal guidelines do not allow for a distinct conclusion section, thus we cannot follow this recommendation.

Appendix

- A great Appendix with lots of supplementary and necessary information to fully understand the methods used.

Thanks.

- In general, I found the appendix order a bit chaotic and suggest ordered bullet points with clear references, e.g. line 103, "see SI" is a bit narrow.

You are right. In fact, journal guidelines require a more structured approach. Each supplementary element is now named Supplementary Figure / Table / Note/ Method / Discussion and referenced accordingly. We also sorted the Supplementary Information per type, which will help in finding references.

- Fig. S5. Additionally (source 28, Haberl et al.) you can consider global building morphology indicators: Biljecki, F., & Chow, Y. S. (2022). Global building morphology indicators. *Computers, Environment and Urban Systems*, 95, 101809.

We don't think that the mentioned paper can be added to the Haberl reference in this context, as the reference provides the detailed math to derive the volume from other measurement units. However, thank you for this paper, the figure with the volume definition is well aligned with our definition of volume, and we added the reference a few sentences earlier when referring to LOD1 blocks.

- Fig. S.1 Pls. mark below the fig. that the flowchart was edited and the original is from Helmut Haberl, Dominik Wiedenhofer, Franz Schug, David Frantz, Doris Virág, Christoph Plutzer, Karin Gruhler, Jakob Lederer, Georg Schiller, Tomer Fishman, Maud Lanau,

Andreas Gattringer, Thomas Kemper, Gang Liu, Hiroki Tanikawa, Sebastian van der Linden, and Patrick Hostert

Environmental Science & Technology 2021 55 (5), 3368-3379

DOI: 10.1021/acs.est.0c05642

Okay, added accordingly.

Further general remarks and suggestions

- Line 102-103: R^2 calculation. You rely on your appendix, page 32 where unit and calculation is explained (per area on country level). Please outline in the text “unit area” because “unit” is not self-explanatory. In this regard Pls. also remark the appendix bullet point or page more precisely as I had to search this information. Pls. explain the noted difference between R^2 0.93 for training and 0.83 for testing

Yes, we apparently overcomplicated this sentence. Thank you for the hint. It now reads as:

“The building and mobility infrastructure stock densities ($t\ km^{-2}$) are highly correlated ($R^2 = 0.88$, $p < 2.2 \times 10^{-16}$, $n = 3,108$, see Supplementary Methods 4)”.

Please note that we removed the split into training and testing as we merely want to assess correlation, and not to obtain a model that could be used to predict one variable from the other.

- Line 124 + 157: You mean “northern Great Planes”, right? North Planes is a city.

Yes! Thank you.

- Line 172 + 204: You could mention the “United Nation Sustainable Development Goals” in the Introduction state of the art as scientific demand, as you rely on climate change in discussion section twice.

Thank you for this suggestion. We included the SDGs in the introduction.

Originality and Significance

In the manuscript an established methodology (DOI: 10.1021/acs.est.0c05642) is used that was set up by some of the authors before. However, you apply and methodologically as well

as spatially extend it to the CONUS area and use multiple calculations to underline scientific significance. A logic way to proof its global significance and continue prior research.

Previous estimations by Sreek et al. or Fishman et al, are met, yet a more deep-going and innovative approach has been done.

Thank you for this evaluation.

The findings also have significance in remote sensing with regard to settlement geography and urban morphology. In these disciplines so far, recent literature has shown that remote sensing urgently needs to be combined more often with socio-economic indicators, e.g.

- Sandborn, A. & Engstrom, R. N. (2016): Determining the Relationship Between Census Data and Spatial Features Derived From

High-Resolution Imagery in Accra, Ghana, IEEE Journal of Selected Topics in Applied Earth Observations and Remote Sensing,

9(5), 1970-1977.

- Taubenböck H, Wurm M, Setiadi N, Gebert N, Roth, A, Strunz G, Birkmann J & Dech S (2009): Integrating Remote Sensing and Social Science – The correlation of urban morphology with socioeconomic parameters. IEEE-CPS Urban Remote Sensing Joint Event (JURSE), Shanghai, China. pp. 7.

Thank you for this positive remark. We absolutely agree. We added the following sentence to the end of the discussion section, but used a more recent reference:

“Beyond providing accurate and spatially explicit high-resolution estimates of material stocks distribution as a key input to socio-economic metabolism research, our study advances urban remote-sensing based research. We here establish a strong link between remote sensing imagery and socio-economic analysis, which has been identified as a strategic research goal⁴⁸. Additional research will need to be done outside the United States, where data availability and reference information is even lower, but where the demand for information is even higher, such that unique urban morphological patterns, e.g., planned vs. unplanned settlements, can have so far unknown effects on material intensity.”

48. Zhu, Z. *et al.* Understanding an urbanizing planet: Strategic directions for remote sensing. *Remote Sens. Environ.* **228**, 164–182 (2019).

Possible points of contact

Your measurements have demonstrated global building datasets. It would be very interesting to use the demonstrated methods to unveil building stock material in informal and poverty areas. Data from these areas about material stocks remain mostly undiscovered by Earth Observation so far. Thus, less obtained poverty areas also play a crucial role in your manuscript (as you mention e.g., Bronx in your article with lowest material intensity and it is well visualized in the web viewer). You respectively mentioned the relevance of the socio-economic findings in the discussion section, line 169, per capita GDP, yet that explanatory power was missing. Why? You could continue research in this area. State of the art literature that contain data barriers between formal and informal areas, rep. poverty areas, e.g.:

Kraff, N. J., Wurm, M., & Taubenböck, H. (2020). The dynamics of poor urban areas-analyzing morphologic transformations across the globe using Earth observation data. *Cities*, 107, 102905.

Dovey, K., van Oostrum, M., Chatterjee, I., & Shafique, T. (2020). Towards a morphogenesis of informal settlements. *Habitat International*, 104, 102240.

Thank you for this suggestion. We agree that there is potential in further research about many aspects that we only briefly touched upon in this study. The Bronx example is very interesting in multiple ways, as the Bronx is also one of the densest areas of our study area. We think that a more detailed and systematic perspective on the relation of socio-economic factors, settlement morphology, and material stock distribution is desirable, but beyond the scope of this study. We expanded our statement about further research and specifically mention this factor:

“Additional research will need to be done outside the United States, where data availability and reference information is even lower, but where the demand for information is even higher, such that unique urban morphological patterns, e.g., planned vs. unplanned settlements, can have so far unknown effects on material intensity.”

Regarding the missing GDP in the multivariate regression, please see our response to your comment on line 169.

References

The number of references is appropriate and covers most state of the art as well as methods and backgrounds. From my perspective missing parts are mentioned above (see Introduction, covering state of the art).

Thank you for this evaluation, we have followed your advice on the state of the art section, please see above.

Thank you!

Thank you for this review.

REVIEWER COMMENTS

Reviewer #1 (Remarks to the Author):

The paper presents a very high level quantification of the material stocks in road and building infrastructure in the continental US. The findings confirm work by others while providing a central dataset for the entire area. The methods used are standard in the field. The authors have satisfactorily addressed my concerns from the previous version in regards to explaining novelty in the context of the existing literature. The findings on the ratios of building mass to transportation mass in different parts of the US are particularly interesting, and I look forward to citing them. I wish the paper had gone further in the implications of its findings, the so what? part. The argument that we should recycle materials from low efficiency stocks is thin. Much can't be moved or recycled without significant down cycling (see concrete). How could this information really be used? That said it is publishable in its current forms as a forward iteration in line with other work in the area.

line 59. Built environment stocks are challenging to reuse except in situ (with some exceptions like asphalt).

Figure 2: Would be helpful to have definitions for what is the line between low rise, low/mid rise, mid rise, high rise etc. What is included in light weight homes other than mobile homes?

line 130. I continue to be unconvinced I should care about people living in biomass dominated areas or not. The paper would benefit from a stronger argument for why this is a factor the reader should be concerned about or how/for what it should be used. For example, people living in areas dominated by built stocks strikes me as a good thing. We should not be so spread all over the land that nothing is natural. We can leave some nature to nature.

line 314: It would be helpful to have a specific section for basements and/or building depth.

Reviewer #2 (Remarks to the Author):

The authors have made many attempts to address concerns raised in the initial review. There are several improvements to the paper, including a visual and insightful quantification and communication of uncertainty. The revised Fig. 2 is impressive and information dense, and I particularly appreciate the addition of Fig. 2c to compare uncertainties for different product types and locations. The authors maybe undersell their contribution when summing up their findings in the first paragraph of the Discussion. I would also suggest to include reference to the large difference in distribution of material intensities for building and mobility infrastructures here, and how they are more weakly or strongly associated with population concentrations. A more detailed comment on this point is found below. The regression results are more straightforward and defensible now, and the controlling for spatial autocorrelation is appreciated.

In the closing Discussion, the authors recommend 'that the built-up environment should be gradually modified to reflect the current and projected patterns of population distribution.' This is a reasonable recommendation, but the suggestions that follow (while sensible), lack acknowledgement of practical challenges associated. I have a more detailed comment on this below. To really test out the environmental implications of spatially explicit population and material stock projections, a separate model would be needed to simulate the flows of materials from hibernating stocks and/or primary extraction and processing, under different assumptions/scenarios of material recovery and modification of existing built environments (including demolition/recovery of underutilized stocks). Ideally this would include consideration of difficult questions such as storage of recovered materials when there is not a current (or closeby) demand, and the impacts associated with recovering materials and transporting them to where they may be needed. That would be a large paper in itself, and I am hopeful that the authors consider that in their future plans.

I find the current manuscript mostly publishable in its current form, although I still have some comments and suggestions on the text and the interpretation/discussion of results, which could still improve the article further. These are summarised below:

- Lines 141-143: another way of phrasing this is that mobility stocks are more evenly dispersed through the country while building stocks are much more sensitive to where populations agglomerate. This is reflected in another way in Fig. 5; because buildings only 'occur' where populations exist, there is less spread in t/cap of building stocks, whereas mobility stocks are ubiquitous and less sensitive to population concentrations, and so areas with low populations (or densities) end up having very high mobility stocks per cap.
- Line 161-162, we don't see the particularly low intensities in urban counties, probably because they are so small as you replied in a comment. Maybe good to clarify that with readers here, as they might read this sentence and then go looking for the low intensities in that region and not be able to really see them.
- Line 163-165, you could optionally add some additional interpretation and explanation of the much wider spatial disparity and wider distribution of material intensities for mobility infrastructures.
- I am a little underwhelmed with the main findings report in lines 174-179. Up to the authors to decide what the most important summary finding are, but personally I also found that the difference in distributions in mobility and buildings infrastructure, and how they vary (or not) with population concentrations to be an interesting outcome of this work. I don't know how novel this result is, but it seems very relevant to the discussions on spatial distributions of human populations and anthropogenic material stocks.
- In addition to the description of highest uncertainty by climate (lines 191-192) would you also add a summary of highest uncertainty by product group (i.e. low rise buildings and local roads)?
- There is a discrepancy between your assessment of whether your numbers are more likely to be underestimating or overestimating material stocks. Compare line 183-184 ('Most potential error sources relating to the employed geodata will mostly result in an underestimation of the total material stock ...') and lines 197-198 ('Hence, our national-scale estimation provides a reasonable estimate, which most likely rather over- than underestimates material stocks.') Can you please make more consistent evaluation, based on your data and interpretation?
- Lines 212-216. Readers might benefit from additional context to know why this is important. You could make an analogy to production vs consumption based environmental footprints for example, where these material intensity estimates here are location-based (allocated to the nighttime populations), rather than use-based (allocated to the populations who use and benefit from the material stocks).
- Ln 235, the reference of metal stocks per cap in electricity infrastructure is helpful. Personally I would find an additional comparison to vehicle stocks more helpful for readers to understand the magnitude of your estimates. A back of the envelope calculation led me to an estimate of 1.6 t/cap of materials in vehicles in the USA. That really helped me to grasp how large your estimate of 391 t/cap of materials in buildings and mobility infrastructure is. Not sure if a good reference is available for that. Interestingly (and maybe less surprisingly), vehicle stocks per cap are also highest in the plains (<https://www.forbes.com/advisor/car-insurance/car-ownership-statistics/#NationalAutomobileDealersAssociation>) but that is perhaps something for another paper.
- 'Spatial' population distribution would help to clarify, on line 272.
- Line 275-277, this sentence is not phrased well.
- Line 281, it is not clear what 'this category' refers to. Growing areas of the US? Densification? Please clarify
- Ln 271-282. You make some understandable recommendations on modifying built environments to better reflect population distributions, reduce (raw) material inputs and associated environmental impacts. Do you have anything to say about practical implementation and challenges for these recommendations? For instance, if thinning an underutilized road network in a low population density location with low projected population growth, what to do with the recovered materials if there is little/no demand for these materials in new construction locally? Would the environmental benefits of displacing raw material inputs be larger than the environmental costs of recovering the materials from

the underutilized road and transporting them to where they will be used? I don't ask for you to answer these questions in this research, but I feel that such concerns are important and relevant enough to be acknowledged here. Perhaps such questions can be tackled in future research.

Reviewer #3 (Remarks to the Author):

Dear authors

Thank you very much for your revised manuscript and the detailed responses.

After having carefully screened all remarks I have noticed a very detailed revision.

Most suggestions have been implemented and discussed properly. Those suggestions where you had decided not to make changes have been argued in a reasonable way.

I have nothing more to add and authorize the publication.

Thank you.

Reviewer #1 (Remarks to the Author):

The paper presents a very high level quantification of the material stocks in road and building infrastructure in the continental US. The findings confirm work by others while providing a central dataset for the entire area. The methods used are standard in the field. The authors have satisfactorily addressed my concerns from the previous version in regards to explaining novelty in the context of the existing literature. The findings on the ratios of building mass to transportation mass in different parts of the US are particularly interesting, and I look forward to citing them. I wish the paper had gone further in the implications of its findings, the so what? part. The argument that we should recycle materials from low efficiency stocks is thin. Much can't be moved or recycled without significant down cycling (see concrete). How could this information really be used? That said it is publishable in its current forms as a forward iteration in line with other work in the area.

We thank the reviewer for helpful comments and suggestions. Please note that high resolution (10m) mappings of all built structures and their material stocks for entire large countries like the USA have not been presented anywhere; so far only relatively small countries like Germany & Austria, as well as local areas/regions have been mapped at comparable thematic or spatial resolution. This study therefore substantially advances the state-of-the-art.

Secondly, we have edited our text throughout to more clearly communicate the potentials of these maps to inform sustainable resource use strategies, drawing on the widely established 9R concept (Potting et al. 2017).

Clearly, re-use / repair / re-purposing of existing stocks is a key strategy to reduce resource use and associated environmental impacts, followed by measures aimed at improved recycling, which requires spatially explicit information to understand the potentials for 'urban' mining of secondary resources stocked in existing built structures and to plan for fine-scaled networks of recycling facilities minimizing transport. While asphalt is often recycled *in-situ*, sand & gravel partially as well, especially in infrastructure re-construction, concrete recycling is a tiny phenomenon yet. Concrete is crushed and together with a bit of new cement is used as concrete again, however usually only for lower quality applications where high-compressive strength is not so important. All other materials require dedicated technical facilities, implying large transport and storage requirements.

Potting, J., Hekkert, M. P., Worrell, E., & Hanemaaijer, A. (2017). Circular economy: measuring innovation in the product chain. Planbureau voor de Leefomgeving, (2544).

line 59. Built environment stocks are challenging to reuse except in situ (with some exceptions like asphalt).

Thank you for pointing this out. Reviewer 2 had similar concerns. Please see our response to their detailed comment, where we have added practical considerations (last comment R2).

Figure 2: Would be helpful to have definitions for what is the line between low rise, low/mid rise, mid rise, high rise etc. What is included in light weight homes other than mobile homes?

This information is given in the Supplementary Information. To increase its findability, we have added references to the corresponding supplementary tables in the caption of Figure 1 (“**Definitions for employed categories are summarized in Supplementary Tables 4, 10, and 13.**”), and have added the height ranges to the figure itself. We believe that Figure 1 is the better place to provide this information as it presents the overview of the employed classes.

line 130. I continue to be unconvinced I should care about people living in biomass dominated areas or not. The paper would benefit from a stronger argument for why this is a factor the reader should be concerned about or how/for what it should be used. For example, people living in areas dominated by built stocks strikes me as a good thing. We should not be so spread all over the land that nothing is natural. We can leave some nature to nature.

We agree with the last statement: areas still dominated by plants should stay dominated by plants. We do not advocate further spreading into nature. In our opinion, it is rather the other way around, i.e., reducing the human impact in rural areas where an disproportionately large mass of artificial materials is located when compared to the relatively small population using these infrastructures, especially in regions characterized by out-migration. When we picture rural areas, we generally expect a lot of plants, but our data clearly suggests that, in many cases, these areas still have more mass in artifacts (roads, and to a lesser extent buildings and other infrastructures), than in plant biomass, which we did not expect, and might hence also be an unexpected insight for others. In these areas, there are several reasons to assume that improving the ratio of plant biomass vs built structures (esp. streets) could have environmental benefits and contribute to climate-change mitigation. Secondly, even in population centers, biomass should exist to mitigate heat stress, improve water management and for mental health. Thus, also in cities, it would be beneficial to increase the ratio of biomass stocks versus built-up stocks. We have emphasized these aspects in the

introduction, results description, and discussion accordingly. We specifically reworded the sentence “This finding suggests that very few people dwell in biomass-dominated ‘green’ landscapes” as we realized it might be misunderstood as a suggestion to spread development into nature.

line 314: It would be helpful to have a specific section for basements and/or building depth.

We have edited the method section as far as space constraints allow us to. The supplementary information contains more detailed documentation, as does the previously published methods article (Haberl et al. 2021). Summarized, we start from information about typical buildings and all materials contained in them, including all underground structures such as basements and foundations. As remote-sensing information only yields data on “above-ground stocks”, we re-estimate this original information on total material stocks per building, into tons / m³ of above ground building volume.

Reviewer #2 (Remarks to the Author):

The authors have made many attempts to address concerns raised in the initial review. There are several improvements to the paper, including a visual and insightful quantification and communication of uncertainty. The revised Fig. 2 is impressive and information dense, and I particularly appreciate the addition of Fig. 2c to compare uncertainties for different product types and locations. The authors maybe undersell their contribution when summing up their findings in the first paragraph of the Discussion. I would also suggest to include reference to the large difference in distribution of material intensities for building and mobility infrastructures here, and how they are more weakly or strongly associated with population concentrations. A more detailed comment on this point is found below. The regression results are more straightforward and defensible now, and the controlling for spatial autocorrelation is appreciated.

Thank you for this positive evaluation. We have addressed your detailed comments below.

In the closing Discussion, the authors recommend ‘that the built-up environment should be gradually modified to reflect the current and projected patterns of population distribution.’ This is a reasonable recommendation, but the suggestions that follow (while sensible), lack acknowledgement of practical challenges associated. I have a more detailed comment on

this below. To really test out the environmental implications of spatially explicit population and material stock projections, a separate model would be needed to simulate the flows of materials from hibernating stocks and/or primary extraction and processing, under different assumptions/scenarios of material recovery and modification of existing built environments (including demolition/recovery of underutilized stocks). Ideally this would include consideration of difficult questions such as storage of recovered materials when there is not a current (or closeby) demand, and the impacts associated with recovering materials and transporting them to where they may be needed. That would be a large paper in itself, and I am hopeful that the authors consider that in their future plans.

Thank you for this remark and suggestions. We agree that fully answering these questions needs follow-up studies linking flows to stocks in a spatially explicit manner, for which our results can be highly useful. In general, we are convinced that our generated dataset holds a lot of value and information that we could not investigate in this specific study, but which would be worth pursuing in the future - either by ourselves or by third parties using our openly shared datasets. Please also see our response to your last comment below.

I find the current manuscript mostly publishable in its current form, although I still have some comments and suggestions on the text and the interpretation/discussion of results, which could still improve the article further. These are summarised below:

- Lines 141-143: another way of phrasing this is that mobility stocks are more evenly dispersed through the country while building stocks are much more sensitive to where populations agglomerate. This is reflected in another way in Fig. 5; because buildings only 'occur' where populations exist, there is less spread in t/cap of building stocks, whereas mobility stocks are ubiquitous and less sensitive to population concentrations, and so areas with low populations (or densities) end up having very high mobility stocks per cap.

Yes, thanks for highlighting these systemic relations; mobility infrastructure also serves agriculture, forestry and various other industrial activities in resource extraction, material processing and trade, therefore it is to be expected that those transport networks are much more spread out so that land can be accessed. As those patterns have not been quantified in a spatially explicit manner for entire countries, we think our results are highly novel and will inspire future work. The mentioned lines still describe absolute masses, though. We believe that it is too early in the manuscript to refer to these points. However, we brought those considerations into the edited discussion section.

- Line 161-162, we don't see the particularly low intensities in urban counties, probably because they are so small as you replied in a comment. Maybe good to clarify that with readers here, as they might read this sentence and then go looking for the low intensities in that region and not be able to really see them.

Following your suggestion, we have clarified this issue, and additionally refer the reader to the source data of this figure where they can find the numeric values of all counties as a table (bold is new):

“In addition, urban counties have particularly low intensities, e.g., the urban centers along the Boston-Washington corridor. **However, note that urban counties are usually small in size, thus they are not easily distinguishable in Fig. 5a, but please refer to the online version of this article where the figure's source data is provided as a table.** “

- Line 163-165, you could optionally add some additional interpretation and explanation of the much wider spatial disparity and wider distribution of material intensities for mobility infrastructures.

Good point. We have provided an explanation for this, along the lines of your previous comment (bold is new):

“The spatial disparity is much higher for the mobility infrastructure (595 t cap⁻¹ measured as interquartile range on the county level) as compared to buildings (110 t cap⁻¹), **which is presumably because most buildings provide shelter to the local population, whereas the existence of a road does not necessarily imply that people are living nearby. Mobility infrastructures also serve agriculture, forestry and various other industrial activities in resource extraction, material processing and trade. Therefore, it is to be expected that those transport networks are much more spread out so that land can be accessed.** “

- I am a little underwhelmed with the main findings report in lines 174-179. Up to the authors to decide what the most important summary finding are, but personally I also found that the difference in distributions in mobility and buildings infrastructure, and how they vary (or not) with population concentrations to be an interesting outcome of this work. I don't know how novel this result is, but it seems very relevant to the discussions on spatial distributions of human populations and anthropogenic material stocks.

Thank you for this suggestion. We agree that this is a main finding of our work. We believe that these points were already included in the previous section, although very compressed. We have expanded the section accordingly and now mention this aspect explicitly. The new section reads as (bold is new):

“Large-scale, high-resolution mappings of societal material stocks are an **increasingly hot** research topic **for sustainability** ⁴⁹. Generating a spatially explicit, high-resolution map of material stocks across the US using a novel workflow **based on various geo- and satellite data**. We found that the distribution of material stocks as well as the share of buildings vs. mobility infrastructure stocks and their relation to plant biomass are highly variable across the CONUS, **and that most people are living where built-up stocks outweigh plant biomass. The rural population predominantly lives in areas with heavier mobility infrastructures while more mass is in buildings where urban populations agglomerate, respectively. Mapping material intensity in t cap⁻¹ revealed different spatial patterns and disparity for buildings and mobility infrastructures. A remarkably high variability and a pronounced spike in intensity occurs along the 100th meridian for the mobility infrastructures, whereas material intensity in buildings is more uniform across the US. The existence of different spatial patterns in both categories suggest that the spatial distribution of these artifacts have different sensitivity to population concentrations and that different socio-economic factors might be involved.** While numerous factors introduce uncertainty along the workflow, our conservative bottom-up estimation of 391 ±18 t cap⁻¹ is in line with previous national-scale estimates (ca. 315-430 t cap⁻¹, Supplementary Figures 9-10) ^{37,50}.”

- In addition to the description of highest uncertainty by climate (lines 191-192) would you also add a summary of highest uncertainty by product group (i.e. low rise buildings and local roads)?

Sure, we have added half a sentence (bold is new):

“We found highest uncertainty in climates that necessitate more reinforced construction due to wet and cold conditions, **as well as in the most abundant categories, i.e., low-rise residential buildings and local roads.** “

- There is a discrepancy between your assessment of whether your numbers are more likely to be underestimating or overestimating material stocks. Compare line 183-184 (‘Most potential error sources relating to the employed geodata will mostly result in an

underestimation of the total material stock ...') and lines 197-198 ('Hence, our national-scale estimation provides a reasonable estimate, which most likely rather over- than underestimates material stocks.') Can you please make more consistent evaluation, based on your data and interpretation?

Thank you very much for noticing this. There are both uncertainties that indicate under- and overestimation. We assume that underestimation terms dominate. However, as the interplay between different components in our workflow is very complex, we have reformulated the corresponding sentences and now formulate this more openly - also in agreement with the section in II. 243-256 (clean revised version).

- Lines 212-216. Readers might benefit from additional context to know why this is important. You could make an analogy to production vs consumption based environmental footprints for example, where these material intensity estimates here are location-based (allocated to the nighttime populations), rather than use-based (allocated to the populations who use and benefit from the material stocks).

Thanks for pointing out this potential similarity to the production- vs consumption-based footprint debate. We have edited slightly to communicate this potential alternative interpretation, which however would require highly detailed auxiliary socio-economic data and complex modelling efforts. The section now reads as (bold is new):

"It is also worth noting that the **localized** material intensity **presented herein** disregards that some built-up structures (partially) serve purposes that are not directly related to the local (nighttime) population. Examples are **mobility infrastructure** connecting distant population centers, **or government facilities, as well as industrial and commercial buildings serving populations located elsewhere. A spatially-explicit understanding how stock dynamics are driven by demand elsewhere therefore constitutes an interesting new research avenue, with similarities to consumption-based environmental footprinting helping to understand distant drivers and responsibilities 45.** "

Arjen Y. Hoekstra, Thomas O. Wiedmann (2014): Humanity's unsustainable environmental footprint. *Science* 344, 1114-1117. DOI: [10.1126/science.1248365](https://doi.org/10.1126/science.1248365)

- Ln 235, the reference of metal stocks per cap in electricity infrastructure is helpful. Personally I would find an additional comparison to vehicle stocks more helpful for readers to understand the magnitude of your estimates. A back of the envelope calculation led me to an estimate of 1.6 t/cap of materials in vehicles in the USA. That really helped me to grasp how

large your estimate of 391 t/cap of materials in buildings and mobility infrastructure is. Not sure if a good reference is available for that. Interestingly (and maybe less surprisingly), vehicle stocks per cap are also highest in the plains (<https://www.forbes.com/advisor/car-insurance/car-ownership-statistics/#NationalAutomobileDealersAssociation>) but that is perhaps something for another paper.

Thanks for this suggestion, we have included a per capita stock estimate for passenger vehicles in the USA from the literature. According to Pauliuk et al. 2021, there are about 0.89 t / cap material stocks in passenger vehicles in the USA. The spatial pattern in the plains is indeed very interesting, and we agree that further investigating this is rather something for future research.

Pauliuk, S., Heeren, N., Berrill, P. et al. Global scenarios of resource and emission savings from material efficiency in residential buildings and cars. *Nat Commun* 12, 5097 (2021). <https://doi.org/10.1038/s41467-021-25300-4>

- ‘Spatial’ population distribution would help to clarify, on line 272.

Agreed, the sentence now reads as (bold is new):

“We recommend that the built-up environment should be gradually modified to reflect the current and projected patterns of the **spatial distribution of** population. “

- Line 275-277, this sentence is not phrased well.

We rephrased the sentence, it now reads as:

“In regions experiencing population growth, re-designing and adapting existing stocks, as well as increasing population density and improved recycling would contribute to reducing primary material extraction and industrial processing, as well as associated environmental impacts and emissions⁵¹. Renaturation could improve wildlife habitats, ecosystem resilience, and help mitigate climate change in areas where stocks are removed or re-designed⁵².”

- Line 281, it is not clear what ‘this category’ refers to. Growing areas of the US? Densification? Please clarify

We specified it accordingly. The sentence now reads as (bold is new):

“[...] where it is to be expected that suburban areas dominated by single-family houses with large carbon footprints ⁴⁶ would fall into this category (**population growth with high material intensity**) and should be focus areas of sustainable densification. “

- Ln 271-282. You make some understandable recommendations on modifying built environments to better reflect population distributions, reduce (raw) material inputs and associated environmental impacts. Do you have anything to say about practical implementation and challenges for these recommendations? For instance, if thinning an underutilized road network in a low population density location with low projected population growth, what to do with the recovered materials if there is little/no demand for these materials in new construction locally? Would the environmental benefits of displacing raw material inputs be larger than the environmental costs of recovering the materials from the underutilized road and transporting them to where they will be used? I don't ask for you to answer these questions in this research, but I feel that such concerns are important and relevant enough to be acknowledged here. Perhaps such questions can be tackled in future research.

Thank you for this comment. You are correct. Our suggestions are rather idealistic in nature, and we have amended the text by listing some practical challenges and clearly noting these as future pathways for highly interdisciplinary research:

“We are aware that such ideas may be confronted with practical challenges. Most importantly, recovery, transportation and processing for recycling involves environmental and economic costs that need to be considered in a spatially explicit context. Our spatially resolved dataset can therefore be helpful for a first order assessment of potential sources in close geographical vicinity to areas where those materials might be reused, although siting decisions for recycling plants would certainly require additional investigation. Also, complete recycling is practically impossible, while current construction standards also limit the amount of recycled materials which can be used. Clearly, this is a complex issue, which requires in-depth and interdisciplinary research in the future, necessitating the development of an integrated geospatial model to simulate stock dynamics and spatial patterns, and associated material flows of recovered materials under different projections of population development and their distribution, as well as policy-relevant scenarios on the re-design of existing stocks, and demand for new structures.”

Reviewer #3 (Remarks to the Author):

Dear authors

Thank you very much for your revised manuscript and the detailed responses.

After having carefully screened all remarks I have noticed a very detailed revision.

Most suggestions have been implemented and discussed properly. Those suggestions where you had decided not to make changes have been argued in a reasonable way.

I have nothing more to add and authorize the publication.

Thank you.

Thank you very much for your positive evaluation!

REVIEWERS' COMMENTS

Reviewer #2 (Remarks to the Author):

Thank you for your revisions. I am satisfied that this article can be published in its current state.